# Neural Markov Controlled SDE: Stochastic Optimization for Continuous-Time Data

**Sung Woo Park**[1]**, Kyungjae Lee**[2]**, Junseok Kwon**[3]
[1,3]School of Computer Science and Engineering, Chung-Ang University, Korea
[1,2,3]Artificial Intelligence Graduate School, Chung-Ang University, Korea
[1]`pswkiki@gmail.com`, [2]`kyungjae.lee@ai.cau.ac.kr`, [3]`jskwon@cau.ac.kr`

## Abstract

We propose a novel probabilistic framework for modeling stochastic dynamics with the rigorous use of stochastic optimal control theory. The proposed model called the neural Markov controlled stochastic differential equation (CSDE) overcomes the fundamental and structural limitations of conventional dynamical models by introducing the following two components: (1) Markov dynamic programming to efficiently train the proposed CSDE and (2) multi-conditional forward-backward losses to provide information for accurate inference and to assure theoretical optimality. We demonstrate that our dynamical model efficiently generates a complex time series in the data space without extra networks while showing comparable performance against existing model-based methods on several datasets.

## 1 Introduction

Recently, there has been interest in using continuous dynamical systems to approximate complex time series. Neural ODE Chen et al. (2018), which opened the way for continuous representation of neural networks, have been widely investigated and thoroughly analyzed by Massaroli et al. (2020). As the stochastic generalization of ODE, Neural SDEs Li et al. (2020) have been proposed by regarding intrinsic stochasticity in data representations (*e.g.*, stock market data). Since the conventional Neural ODE/SDEs only utilize the initial information of trajectories when propagating dynamics, modelling complex time-series with naive Neural ODE/SDEs has been regarded as inefficient and undesirable choices, as pointed out by Kidger et al. (2020).

To address these problems, Rubanova et al. (2019) presented an auto-regressive model to generalize recurrent neural networks (RNNs) to have continuous hidden dynamics with neural ODE. Furthermore, Chen et al. (2018) proposed an encoder-decoder structure with Neural ODE in the latent space to reconstruct/predict complex data representation. Although the aforementioned approaches produce remarkable results, they focus on suggesting additional probabilistic structures rather than improving the learnability of the Neural ODE model itself. Compared to aforementioned approaches, we focus on solving the fundamental issues of Neural ODE/SDEs. First, we raise two important questions.

**Q1)** *How can we construct an efficient network architecture for Neural ODE/SDE models that do not require additional recurrent networks to model complex time series?*

**Q2)** *How can we train Neural ODE/SDEs that can utilize richer information of observed sequences to accurately generate complex time series?*

As SDEs can be posed as stochastic generalizations of ODEs, we focus on a stochastic framework and adopt the *stochastic optimal control theory* as our primary analysis tool for the rigorous and systematic analysis of the aforementioned problems. Keeping this in mind, the contributions of our paper are to answer the above two questions.

**A1) Novel probabilistic framework for stochastic dynamics.** We propose a novel *neural controlled stochastic differential equation* (CSDE) to model the complex stochastic time series, where multiple control agents are defined to construct local dynamics in their own private temporal states. With this property, the proposed CSDE incorporates Markov dynamic programming, enables our model to directly infer the **complex trajectory on data space** rather than the latent space without any extra network (*e.g.*, encoders/decoders), and shows remarkable efficiency compared to existing methods.

**A2) Novel conditional losses.** We introduce a novel Markov forward conditional (MFcond) loss to utilize multi-conditioned dynamics instead of the conventional dynamics determined by partial initial conditions. The proposed MFcond loss makes our method to model the complex information of

time-series data. To impose regularization and to ensure the optimality of control agents, we also suggest a novel Markov backward conditional (MBcond) loss.

## 2 RELATED WORK

**ODE As a Latent Probabilistic Model.** Rubanova et al. (2019) suggested an ODE-RNN by combining RNN with the latent dynamics induced by the Neural ODE. To deal with irregular time-stamps, exponential-decaying of the hidden states was also discussed by Che et al. (2018). De Brouwer et al. (2019) assumed that the observations are sampled from the stochastic dynamics induced from SDEs and introduced GRU-ODE to approximate the observed stochastic time series.

**SDE As a Latent Probabilistic Model.** Liu et al. (2021) incorporated Neural SDEs with recurrent models as a primary probabilistic dynamical model to generate stochastic continuous-time latent variables. While this SDE model could describe the stochastic dynamics on the latent space with recurrent structures (*e.g.*, RNN encoder/decoder), it required a whole sequence of historical observations as inputs to the model. Unfortunately, this type of formulation leads to non-Markov types of SDEs, which makes it difficult to analyze the probabilistic characteristics of the dynamics. Unlike this model, we focus on the Markov SDEs while maintaining identical objectives.

**Neural CDE and RDE.** Kidger et al. (2020) proposed a data-driven neural controlled differential equation called Neural CDE to incorporate a rough-path analysis theory and model complex time series. Morrill et al. (2021) extended the rough-path theory with a Neural RDE to deal with the continuous time series over long time.

**Generative SDE Models.** Recently, Kidger et al. (2021) suggested SDE-based generative adversarial networks (GANs). Park et al. (2021) utilized the temporal conditional Wasserstein distance to construct GANs for time-series generation.

**Please refer to Appendix A.1 for additional discussion on related works.**

## 3 MARKOV NEURAL CONTROLLED SDE

In Section 3.1, we introduce a novel SDE model that considers temporally private agents. In Section 3.2, we propose the Markov-DP-TP framework to efficiently solve the stochastic optimal control problem with the proposed neural SDE model. Finally, we suggest novel Markov conditional forward and backward losses in Section 3.3 and 3.4, respectively. In the Appendix, we provided the detailed technical definitions.

### 3.1 CONTROLLED STOCHASTIC DIFFERENTIAL EQUATIONS

The basic object of our interest is a controlled $\mathcal{F}_t$-adapted process $X_t^\alpha$ with multiple control agents $\alpha = \{\alpha^1, \cdots, \alpha^M\} \in \mathbb{A}$ where $\mathbb{A}$ denotes the set of admissible control agents. In particular, the stochastic process $X_t^\alpha$ is defined as a solution to the following CSDE:

$$dX_t^\alpha = \sum_{i=1}^{M} w_i(t) b^i \left(t, X_t^\alpha, \alpha^i\right) dt + \sum_{i=1}^{M} w_i(t) \sigma^i \left(t, X_t^\alpha, \alpha^i\right) dW_t, \tag{1}$$

where $b$ and $\sigma : [0, T] \times \mathbb{R}^d \times \mathbb{A} \to \mathbb{R}^d$ are the drift and diffusion functions, respectively. Each control agent $\alpha^i : [0, T] \times \mathbb{R}^d \times \mathbb{R}^m, \alpha^i = \alpha^i(t, X_t; \theta^i), \forall 1 \le i \le M$ is defined as a Markov closed-loop feedback control, which is parameterized by the neural network $\theta^i$. While every agent is defined as a closed-loop feedback-type Carmona (2016b), the solution to the CSDE above, $X_t^\alpha$, is the *Markov process*, which means that process $X_t^\alpha$ is propagated using the information of the current state.

Let $\mathbb{T} = \{t_k\}_{1 \le k \le N}$ be a set of ordered times[1] such that $0 = t_1 < \cdots < t_k < t_l < \cdots < t_N = T$. The set of functions $\{w_i(t)\}_{1 \le i \le M}$ is defined as an indicator function on the intervals, $w_i(t) = \mathbf{1}_{t_k \le t \le t_l}$ with predetermined starting/ending points $t_k, t_l$ in $\mathbb{T}$. We call this function **temporal privacy (TP)** because it represents each agent's attention on different sub-intervals. Overall, in (1), the stochastic process $X_t^\alpha$ is propagated by summing $M$-number of individual agent's weighted attentions $\left\{\sum^M w_i b^i(\cdot, \cdot, \alpha^i), \sum^M w_i \sigma^i(\cdot, \cdot, \alpha^i)\right\}$. To understand the behavior of the proposed CSDE more deeply, we consider the following detailed example:

---

[1] The time interval $dt \approx \Delta_t = |t_k - t_l|$ for any $k, l$ can be set regularly/irregularly in our method.

**Role of Temporal Privacy.** We define $w_r(s) = \mathbf{1}_{t \leq s \leq u}$, $t, u \in \mathbb{T}$ with $r \leq M$. Then, $X_u^\alpha$ in (1) given $X_t$ at an interval $[t, u]$ can be equivalently rewritten in the integration form:

$$
\begin{aligned}
X_u^{\alpha=[\alpha^1, \cdots, \alpha^M]} &= X_t^\alpha + \int_t^u \sum_i^M w_i(s) b^i(s, X_s^\alpha, \alpha^i) ds + \int_t^u \sum_i^M w_i(s) \sigma^i(s, X_s^\alpha, \alpha^i) dW_s \\
&= X_t^\alpha + \int_{w_r(s)} b^r(s, X_s^\alpha, \alpha^r) ds + \int_{w_r(s)} \sigma^r(s, X_t^\alpha, \alpha^r) dW_s = X_u^{\alpha^r}.
\end{aligned}
\tag{2}
$$

In (2), the activated control agent to evaluate the stochastic process $X_u^\alpha$ for the interval $[t, u]$ is only $\alpha^r$ (*i.e.*, $X_u^\alpha = X_u^{\alpha^r}$) owing to the definition of the weighting function $w_{(\cdot)}(t)$. This means that the remaining control agents $\{\alpha^j\}_{j \neq r}$ are not used for the evaluation of the stochastic process in the sub-interval $[t, u]$. While each agent $\alpha^i$ is activated at its own private sub-interval, this leads our method to adopt *dynamic programming (DP)* to train Neural CSDEs in the form of (1). In this paper, we aim to solve the optimal control problem via DP with multiple agents, where each agent specializes in solving a particular sub-problem in its private interval.

## 3.2 MARKOV DYNAMIC PROGRAMMING PRINCIPLES

The *dynamic programming principle* is one of the fundamental philosophies for dealing with stochastic optimal control problems. Its basic idea is to consider a family of sub-problems with different initial times/states and establish the relation among the sub-problems to systemically solve them. Using the mathematical property of the proposed CSDE with TP, we present an efficient learning strategy to solve stochastic optimal control problems via Markov dynamic programming (Markov-DP).

In this paper, we aim to solve the stochastic optimal control problem by training control agents $\alpha = [\alpha^1, \cdots, \alpha^M]$ and minimizing the *cost functional* $J(t, X_t^\alpha) : [0, T] \times \mathbb{R}^d \to \mathbb{R}^+$:

$$
J(t, X_t^\alpha) = \mathbb{E}\left[ \int_t^T l(s, X_s^\alpha) ds + \Psi(X_T^\alpha) \Big| \mathcal{F}_t \right] = \mathbb{E}\left[ \int_t^u l(s, X_s^\alpha) ds + J(u, X_u^\alpha) \Big| X_t^\alpha \right], \tag{3}
$$

where $l : [0, T] \times \mathbb{R}^d \to \mathbb{R}^+$ is the *running cost* (*e.g.*, $L_2$ loss) that computes the discrepancy between the propagated process $X_t^\alpha$ and the observed data point $y_t$ at each time $t$, $\Psi(X_T^\alpha) : \mathbb{R}^d \to \mathbb{R}^+$ is the *terminal cost* that estimates the discrepancy between the terminal state and the data $y_T$. To evaluate the cost functional $J(t, X_t^\alpha)$ at time $t$ with control agents $\alpha$, the running cost is integrated over the time interval $[t, T]$ conditioned on filtration $\mathcal{F}_t$. Note that the expectation conditioned on $\mathcal{F}_t$ in (3) can be substituted to the expectation conditioned on $X_t^\alpha$ in light of the Markov property presented in Section A.2, and the cost functional at time $t$ only depends on the current state of the process $X_t^\alpha$.

**Markov-DP with Temporal Privacy.** By combining the tower property of the conditional expectations with the dynamic programming principle and Itô's formula (Oksendal (1992)), one can show that a minimization problem can be recursively decomposed into sub-problems owing to the property of TP in our proposed CSDE:

$$
\begin{aligned}
V(t, X_t^\alpha) &\triangleq \inf_\alpha J(t, X_t^\alpha) = \underbrace{\inf_\alpha \mathbb{E}\left[ \int_t^u l(s, X_s^\alpha) ds + J(u, X_u^\alpha) \Big| X_t^\alpha \right]}_{(\mathbf{A})} \\
&= \underbrace{\inf_{\alpha^r} \mathbb{E}\left[ \int_t^u l(s, X_s^\alpha) ds \Big| X_t^\alpha \right]}_{(\mathbf{B})} + \underbrace{\inf_{\alpha^{(-r)}} \mathbb{E}[J(u, X_u^\alpha) | X_t^\alpha]}_{(\mathbf{B'})},
\end{aligned}
\tag{4}
$$

where $V$ is an optimal cost functional (*i.e.*, value function), $\alpha^r$ denotes the $r$-th control agent, and $\alpha^{(-r)} = [\alpha_1, \cdots, 0, \cdots, \alpha_M]$ indicates the set of remaining agents (the $r$-th component is zero). In (4), the minimization problem $(\mathbf{A})$ over $\alpha$ is divided into two sub-problems using the dynamic programming principle, which are $(\mathbf{B})$ and $(\mathbf{B'})$. Because the minimization problem $(\mathbf{B})$ is only dependent on the control agent $\alpha^r$ parameterized by the neural network $\theta^r$, we compute the gradient descent of $\theta^r$ to solve the sub-problem $(\mathbf{B})$:

$$
\theta_{k+1}^r = \theta_k^r - \frac{\partial}{\partial \theta^r} \mathbb{E}\left[ \int_{\{s:w_r(s)=1\}} l\left( s, X_s^{\alpha^r(\cdot, \cdot, \theta_k^r)} \right) ds \Big| X_t^\alpha \right], \tag{5}
$$

where $w_r(s) = \mathbf{1}_{t \leq s \leq u}$ is the TP function at an interval $[t, u]$ and $k$ is the index for the learning iterations. In (5), the $r$-th control agent $\alpha^r$ minimizes the cost functional using the gradient descent scheme at its own temporal sub-interval. As the remaining sub-problem **(B')** over agents $\alpha^{(-r)}$ can also be recursively decomposed into smaller sub-problems using the dynamic programming principle, the original problem **(A)** is solved separately with $M$-number of control agents $\alpha = \{\alpha^1, \cdots, \alpha^M\}$ with the $M$-number of gradient descent schemes. This indicates that we can obtain the set of agents $\alpha_\star = \{\alpha^i(\cdot, \cdot; \theta^i_\star)\}$ by collecting individual optimal agents with sub-problems.

In this paper, we combine the Markov-DP with $M$ gradient descent schemes in (5) and CSDE with TP in (1) and introduce a novel *Markov-DP-TP framework*. In the numerical experiments in Section 4.4, we show that the proposed Markov-DP-TP framework *remarkably increases the model efficiency* compared to conventional non-DP naive approaches, which makes our method directly model the complex time series in the data space. However, despite the improvements with our novel Markov-DP-TP framework, there exist remaining practical/theoretical issues that should be addressed to solve the optimal control problem with complex datasets.

1) **Conditional Dependency.** The main practical issue in implementing the Markov-DP-TP framework is that explicit conditional states are not given, *e.g.*, $X^\alpha_t$ in (5). As different initial/terminal conditions of SDE lead to totally different behaviors of induced dynamics, well-designed conditional information is a crucial factor in training the Neural CSDE for specific applications. In Section 3.3, we introduce the **Markov Forward conditional (MFcond) loss** to train the Neural CSDE with well-posed conditional information that ensures accurate network predictions.

2) **Theoretical Optimality.** In the optimal control theory, there are well-known partial differential equations called Hamiltonian-Jacobi-Bellman (HJB) equations, which assure the theoretical optimality of control agents. If the control agents can solve the HJB equation, the proposed method attains the optimal state $V_t(X^\alpha_t) = \inf_\alpha J_t(X^\alpha_t) = J_t(X^{\alpha_\star}_t)$. However, the optimal agents $\alpha_\star$ of the proposed CSDE with gradient descent are not generally equivalent to the solution to the HJB equation. In Section 3.4, we propose the **Markov Backward conditional (MBcond) loss** to assure the optimality of control agents and to provide information in backward dynamics for regularization.

### 3.3 MARKOV FORWARD CONDITION

In this section, we first raise the important question: *Why is the well-posed conditional estimation in cost functional important to accurately train Neural SDE (CSDE) models?* To elucidate the importance of this question, we consider the following minimization problem with the cost functional with naive partial information:

$$\inf_\alpha \mathcal{L}(\alpha) = \inf_\alpha \mathbb{E}_{y_0} \left[ \int_0^T l(s, X^\alpha_s) ds + \Psi(X^\alpha_T) \middle| X_0 = y_0 \right],\qquad(6)$$

where $\mathbf{y}_{(\cdot)} = \{y_t\}_{t \in [0,T]}$ denotes a set of observed data, and $y_0$ is the initial data at time $t = 0$. In (6), the conditional expectation is taken to the single initial state $X_0 = y_0$, and the control agents minimize the accumulated losses using this partial information. As pointed out by Kidger et al. (2020), this naive cost functional causes a problem when dealing with high-dimensional complex datasets. This is because the Neural CSDE should disentangle the inherent latent information of *complex high-dimensional data* to generate accurate results, but the control agents are trained with only the restrictive and partial information of the observed data (*i.e.*, initial condition $X_0 = y_0$). To solve this problem, we introduce a novel loss function called the **MFcond loss** that can fully exploit the information of the given observed data $\mathbf{y}_{(\cdot)}$, while keeping the Markov structure of $X^\alpha_t$:

**Definition 1.** *(MFcond loss) We define the prediction operator $\mathcal{T}^\alpha_{s,t}$ as follows, for $s < t$,*

$$\mathcal{T}^\alpha_{s,t} := \frac{1}{|I(s,t)|} \sum_{m \in I(s,t)} \left[ X^\alpha_{t_m} + \int_{t_m}^t \sum_{i=1}^M w_i b^i(u, X^\alpha_u, \alpha) du + \int_{t_m}^t \sum_{i=1}^M w_i \sigma^i(u, X^\alpha_u, \alpha) dW^{(m)}_u \middle| X^\alpha_{t_m} = y_{t_m} \right],$$
$$(7)$$

*where $I(s,t) := \{m : s \leq t_m < t\}$, $|I(s,t)|$ is the cardinality of $I(s,t)$, and $\{W^{(m)}_u\}_{m \in I(s,t)}$ denotes the Wiener processes with respect to time $u$. Let us define a random stopping time $\tau_s$ such that $\tau_s := \inf_t \{t : l(t, \mathcal{T}^\alpha_{s,t}) > \epsilon\}$ for the pre-determined threshold $\epsilon$[2]. Then, we can define the MFcond loss with the stopping time $\tau_{(\cdot)}$, as follows:*

---

[2]Please refer to Appendix (A.7) for detailed information

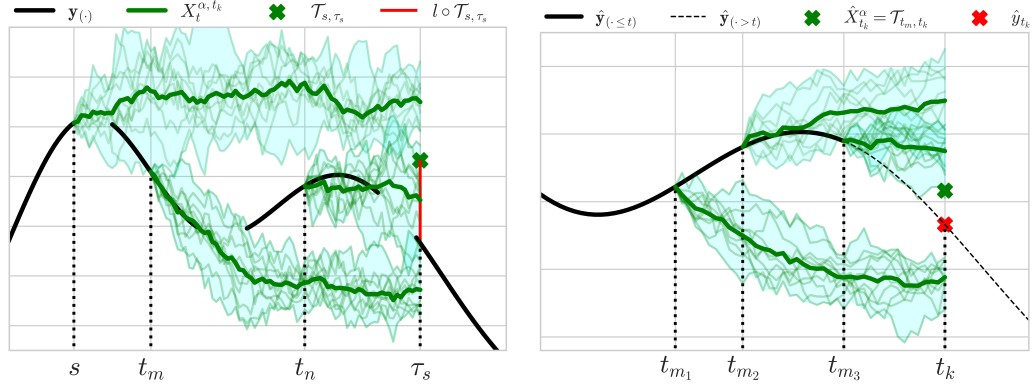

(a) Network Training with the MFcond loss          (b) Network Inference

Figure 1: **Conceptual illustration of the MFcond loss and network inference**. (a) The bold green lines indicate a single trajectory of the proposed CSDE, while other trajectories are shown in shaded regions. The MFcond loss estimates the conditioned losses $l \circ \mathcal{T}_{s,\tau_s}^\alpha$ (*e.g.*, red vertical lines) at $\tau_s$. The empty parts of the black lines indicate the irregularly-sampled (unobserved) data representation. (b) The average decision $\hat{X}_{t_k}^\alpha$ (*e.g.*, green cross) is estimated to approximate the true test data $\hat{y}_{t_k}$ (*e.g.*, red cross) using multiple past observations. This figure shows the failure case as process $\hat{X}_{t_k}^\alpha$ hardly approximates $\hat{y}_{t_k}$.

$$\mathcal{L}_f\left(\alpha, \mathbf{y}_{(\cdot)}\right) = \mathbb{E}_{\mathbf{y}_{(\cdot)}}\left[\int_t^T l\left(\tau_s, \mathcal{T}_{s,\tau_s}^\alpha\right)\chi(s)ds + \Psi(X_T^\alpha)\right], \tag{8}$$

where $\chi(s)$ is an indicator function that produces values at the observed time (*i.e.*, $\chi(s) = 1$ if $y_s$ is observed at $s$; otherwise, $\chi(s) = 0$). This function is used to consider the irregularly sampled data points. In (8), naive running cost $l$ of (6) is replaced with $l \circ \mathcal{T}_{s,\tau_s}^\alpha$, in which the MFcond loss recursively accumulates the expected future losses $l \circ \mathcal{T}_{s,\tau_s}^\alpha$ conditioned on multiple observations.

At each time $s$, stopping time $\tau_s$ decides the future time to stop the CSDE propagation by determining if the accumulated losses are larger than the predetermined threshold $\epsilon$ or not. While the proposed loss requires a set of multiple conditions on the Markov process $X_t^\alpha$ to train control agents, information is utilized to generate time-series data, and complex dynamics can be expressed. A conceptual illustration of the proposed MFcond loss is shown in Figure 1-(a).

The main idea of our MFcond loss in (8) is to minimize the differences between the future estimations $X_u^{\alpha,s}$ for any given $s \leq u$. In other words, the proposed CSDE is trained to generate an *identical future estimation of $X_u^\alpha$ given any past initial conditions $X_{(\cdot)}^\alpha = y_{(\cdot)}$*, *i.e.*, $(X_u^{\alpha,s} \approx X_u^{\alpha,t}, \forall s \leq t \leq u)$ to estimate network inference with multiple conditions in the test time. This idea is used to introduce a novel inference procedure to overcome the raised issues on the partial information.

**Network Inference.** Let $\{y_{t_m}\}$ be the observed data sequences until the current time $t$ in the test dataset. Our objective is to predict the future points $\{\hat{y}_{t_k}\}, (t_m \leq t < t_k)$. Our model generates the stochastic estimation $\hat{X}_{t_k}$ to approximate $\hat{y}_{t_k}$ at a future time given multiple initial conditions $\hat{y}_{t_m}$:

---
**Network Inference**

$$\hat{y}_{t_k} \approx \hat{X}_{t_k}^\alpha = \mathcal{T}_{t_m,t_k}^\alpha =$$
$$\frac{1}{|I|}\sum_{s \in I(t_m,t_k)}\left[X_{t_s}^\alpha + \sum_{i=1}^M \int_{t_s}^{t_k} w_i b^i(t, X_t^\alpha, \alpha^i)dt + \sum_{i=1}^M \int_{t_s}^{t_k} w_i \sigma^i(t, X_t^\alpha, \alpha^i)dW_t^{(s)}\bigg| X_{t_s}^\alpha = \hat{y}_{t_s}\right] \tag{9}$$

---

In (9), each control agent makes decisions on its specialized temporal state and collaborates to generate a stochastic conditional estimation $\hat{X}_{t_k}^\alpha$ and approximate $\hat{y}_{t_k}$. As our MFcond loss induces identical estimations $X_{t_k}^{\alpha,\hat{y}_{t_m}}$ for any $t_m$, $\hat{X}_{t_k}^\alpha$ utilizes multiple conditions $\{\hat{y}_{t_m}\}$ and fully exploits the past information to predict/estimate future values. A conceptual illustration of the network

inference is shown in Figure 1-(b). While the proposed inference mechanism utilizes *enlarged information*[3] compared to a single initial condition, it can model the complex time-series data.

If the control agents are trained with the naive cost functional, the terminal states $X_u^{\alpha,s}$ (conditioned on initial state $X_s = y_s$) and $X_u^{\alpha,t}$ (conditioned on initial state $X_t = y_t$) are largely different, which causes problems when we generate complex time-series data during the test time, whereas our inference mechanism introduced in (9) utilizes averaged multi-decisions $X_{t_k}^{\alpha}$ given different initial conditions. Thus, the MFcond loss is essential for utilizing the proposed inference procedure.

Unlike the dynamical auto-regressive probabilistic models (*e.g.*, ODE-RNNs) that encode whole (or partial) data sequences, as shown in (1), the proposed Markovian CSDE model only uses the current observation to propagate stochastic dynamics. An additional inference mechanism coordinates the multi-conditioned trajectories to utilize information and produces complex time series.

### 3.4 Markov Backward Condition

In the previous section, we suggested the Markov forward conditional loss that exploits the entire information of time-series data to generate accurate results. Aside from its empirical benefits to some applications, no theoretical/empirical optimality of (4) is assured by minimizing the MFcond loss in general. To tackle this problem, in this section, we further introduce the additional stochastic dynamics relating optimality of proposed CSDE-TP.

Let us define the auxiliary process $Z_t = V(t, X_t^{\alpha_\star})$ with a value function $V$, where $\alpha_\star$ denotes the optimal control agents. Subsequently, we consider the following forward-backward stochastic differential equations (FBSDEs):

$$
\boxed{
\begin{array}{l}
\textbf{FBSDEs} \\[4pt]
(X_t^{\alpha_\star}, Z_t) = \begin{cases} dX_t^{\alpha_\star} = \sum_{i=1}^M w_i b^i(t, X_t^{\alpha_\star}, \alpha_\star^i)dt + \sum_{i=1}^M w_i \sigma^i(t, X_t^{\alpha_\star}, \alpha_\star^i)dW_t \\ dZ_t = -l(s, X_t^{\alpha_\star})dt + \sum_{i=1}^M \nabla V(t, X_t)w_i \sigma^i(t, X_t^{\alpha_\star}, \alpha_\star^i)dW_t \\ Z_T = \Psi(X_T^{\alpha_\star}) \end{cases} \quad (10)
\end{array}
}
$$

The first SDE (*i.e.*, $X_t^{\alpha_\star}$) called the *forward SDE* has an identical form of (1) and propagates stochastic evaluation in the forward direction with optimal control agents. The second SDE (*i.e.*, $Z_t$) called *backward SDE* recursively subtracts the running cost from the terminal state $\Psi(X_T^{\alpha_\star})$ in the backward direction using forward estimations $X_t^{\alpha_\star}$ and cancels the effect of martingales in the diffusion term. We utilize the property of backward dynamics $Z_t$ to train the control agents for the following reasons.

**1) Backward Multi-conditions.** Like the MFcond loss with multi-conditions in the forward direction, we want to provide additional information to backward dynamics to train the control agents.

**2) Approximated Solution of HJBE.** The auxiliary process $Z_t$ gives the theoretical optimality for control agents related to the HJB equation based on the results developed in Yong & Zhou (1999); Pardoux & Tang (1999), where the process $Z_t = V(t, \cdot)$ admits a solution of the HJB equation in (11) and induces an optimal solution for the minimization problem $\inf_\alpha J$ in (4).

$$
\frac{\partial V(t,x)}{\partial t} + \frac{1}{2}\mathbf{Tr}[\sigma^T \sigma(t,x,\alpha^\star)\nabla^2 V(t,x)] + \nabla V(t,x)^T b(t,x,\alpha^\star) + l(t,x) = 0, \quad (11)
$$

where $V(T,x) = \Psi(x)$. In (11), we want to approximate $Z_t$ using control agents for optimality. However, the process $Z_t$ requires optimal control agents $\alpha_\star$ that cannot be obtained during the training time. To overcome this problem, we approximate the auxiliary process $Z_t$ with $Z_t^\alpha$ parameterized by neural control agents $\alpha(\cdot,\cdot,\theta)$, which is defined as the modified version of $Z_t$. In particular, $Z_t$ can be expressed in the following integral form:

$$
Z_t^\alpha = \Psi(X_T^\alpha) - \int_T^t \sum_i^M w_i(s)l(s, X_s^\alpha)ds + \int_T^t \sum_i^M w_i(s)\sigma^i(s, X_s^\alpha, \alpha^i)\nabla J(s, X_s^\alpha)dW_s^T, \quad (12)
$$

where $J$ is the cost functional defined in (3), and $\nabla J$ denotes the gradient of the cost functional with respect to its spatial axis. Using the proposed process $Z_t^\alpha$, we introduce a novel loss function called the **MBcond loss** to satisfy the two objectives discussed above.

---

[3]Please refer to detailed explanation in Appendix A.3.

---

**Algorithm 1** Neural Markov CSDE-TP

---

**Require:** $\gamma = 0.95$,
    **for** $k = 1$ to $K$ (*i.e.*, the total number of training iterations) **do**
        **1) Simulate forward controlled SDE with Markov control agents**
        **1-1)** $dX_t^{\alpha_k} = \sum_{i=1}^{M} w_i b^i(t, X_t^{\alpha_k}, \alpha_k^i)dt + \sum_{i=1}^{M} w_i \sigma^i(t, X_t^{\alpha_k}, \alpha_k^i)dW_t$
        **1-2)** Evaluate each decision of control agents $\alpha_k^i = \alpha_k^i(t, X_t^{\alpha_k}; \theta_k^i)$
        **1-3)** Compute the MFcond loss for $M$-control agents $\{\mathcal{L}_f(\alpha_k^i(\cdot, \cdot, \theta_k^i))\}$ with stopping time $\tau_{(\cdot)}$
        **1-4)** Update threshold $\epsilon$ for random stopping time, $\epsilon_{k+1} \leftarrow \frac{1}{2} \max l\left(t, \mathcal{T}_{s,t}^{\alpha_k}(y_s)\right)$

        **2) Simulate backward controlled SDE**
        **2-1)** $dZ_t^{\alpha_k} = -\sum_i^M w_i l(s, X_t^{\alpha_k})dt + \sum_{i=1}^M \nabla J(t, X_t^{\alpha_k}) w_i \sigma^i dW_t$
        **2-2)** Evaluate the MBcond loss for $M$-control agents, $\{\mathcal{L}_b(\alpha_k^i(\cdot, \cdot, \theta_k^i))\}_{1 \leq i \leq M}$

        **3) Update control agents with Markov-DP**
        **3-1)** $\theta_{k+1}^i = \theta_k^i - \gamma \nabla_{\theta^i} \mathcal{L}_f(\alpha^i(\cdot, \cdot, \theta_k^i)) - (1 - \gamma)\nabla_{\theta^i} \mathcal{L}_b(\alpha^i(\cdot, \cdot, \theta_k^i))$
    **end for**

---

**Definition 2.** *(MBcond loss) Let us define the auxiliary process $Z_t^\alpha$ as the solution to (12). Then, the MBcond loss can be defined as follows:*

$$\mathcal{L}_b(\alpha) = \mathbb{E}_{\mathbf{y}_{(\cdot)}, t \in [0,T]} \left[ |Z_t^\alpha|^2 \Big| X_t = y_t \right]. \tag{13}$$

Theoretically, if we optimize the MBcond loss (13) according to the proposed backward dynamics $Z_t^\alpha$, the PDE reformulation of backward dynamics, called *Non-linear Feynmann-Kac*, have the identical solution[4] to HJB equation in (11). Thus, our method can attain the optimal solution of original problem posed in section 3.2.

Intuitively saying, one can show that the MBcond loss is equivalent to the reformulation of the minimization problems in (4) using Itô's formula. Thus, solving the minimization problem $\inf_\alpha \mathcal{L}_b$ induces an identical effect to solve the original problem $\inf_\alpha J$. The only difference is that we utilize multiple conditions to provide conditional information on the backward dynamics $Z_t^\alpha$ for the *regularization of control agents* trained with forward conditional dynamics and to impose constraints on control agents, which induces an approximated solution to the HJB equation.

### 3.5 OBJECTIVE FUNCTION

In this section, we describe the overall training procedure, which incorporates all the proposed components (*i.e.*, Markov-DP with CSDE-TP, MFcond loss, and MBcond loss) as follows:

$$\inf_\alpha \underbrace{\mathcal{L}(\alpha)}_{\text{MFBcond}} = \inf_{\alpha = [\alpha^1, \cdots, \alpha^M]} \underbrace{\gamma \mathcal{L}_f(\alpha)}_{\text{MFcond}} + \underbrace{(1 - \gamma)\mathcal{L}_b(\alpha)}_{\text{MBcond}}$$

$$\overset{\text{CSDE-TP}}{\approx} \sum_i^M \inf_{\alpha^i} \gamma \mathcal{L}_f([\alpha^i, \alpha^{(-i)}]) + \left[ (1 - \gamma)\mathcal{L}_b([\alpha^i, \alpha^{(-i)}]) \right], \tag{14}$$

where $\mathcal{L}_f$ and $\mathcal{L}_b$ are defined in (8) and (13), respectively, and $\gamma$ is a balancing hyperparameter. In (14), the control agents $\alpha = [\alpha^1, \cdots, \alpha^M]$ are trained with a convex combination of MFcond and MBcond losses. Using the property of CSDE-TP with Markov-DP, the original problem is approximated with the collection of $M$ sub-problems, and each control agent is separately trained with $M$ gradient descent schemes. Algorithm 1 describes the detailed procedure of our method.

## 4 EXPERIMENTS

**Network structure of control agents.** The neural network structure for each agent control consists of 2-layers of fully-connected layers, where each module has 128 latent dimensions. For the activation units, we used the specialized module `LipSwish`, Chen et al. (2019); Kidger et al. (2021), to stabilize the FBSDEs during training. Please refer to Appendix A.6 for detailed information on the network architecture.

**Datasets.** For the evaluations, we used PhysioNet, Speech Commands, Beijing Air-Quality, and S&P-500 Stock Market datasets. Refer to Appendix A.5 for data statistics and prepossessing procedures.

---

[4]Please refer to Appendix A.4 for the discussion on theoretical optimality induced by the MBcond loss.

## 4.1 TIME-SERIES DATA RECONSTRUCTION

Table 1: **Evaluation of reconstruction tasks on the PhysioNet/Speech Commands datasets.**

| Methods | PhysioNet | | Speech Commands | |
| --- | --- | --- | --- | --- |
| | MSE $(\times 10^{-1})\downarrow$ | NLL $(\times 10^2)\uparrow$ | MSE $(\times 10^0)\downarrow$ | NLL $(\times 10^3)\uparrow$ |
| RNN-VAE | 6.655 | $-33.238$ | 0.984 | $-4.918$ |
| ODE$^2$VAE | 5.523 | $-24.195$ | 0.723 | $-3.763$ |
| Latent SDE (ODE-Enc) | 2.403 | $-11.978$ | 0.841 | $-4.226$ |
| Latent SDE (RNN-Enc) | 2.418 | $-12.051$ | 0.851 | $-4.250$ |
| Latent ODE (ODE-Enc) | 2.395 | $-11.933$ | 0.851 | $-4.251$ |
| Latent ODE (RNN-Enc) | 2.375 | $-11.840$ | 0.880 | $-4.402$ |
| RNN-Decay | 1.431 | $-7.119$ | 0.865 | $-4.323$ |
| ODE-RNN | 1.311 | $-6.517$ | 0.697 | $-3.479$ |
| GRU-D | 1.424 | $-7.083$ | 0.838 | $-4.116$ |
| mTAND | 0.887 | $-4.396$ | 0.581 | $-2.902$ |
| CSDE-TP | **0.755** | **$-3.667$** | **0.435** | **$-2.172$** |
| CSDE-TP-MV | **0.728** | **$-3.605$** | **0.423** | **$-2.109$** |

In this experiment, we compared our model against baseline dynamic models: [Latent ODE, Chen et al. (2018)], [Latent SDE, Li et al. (2020)], [ODE-RNN, Rubanova et al. (2019)], [GRU-D, Che et al. (2018)], [mTAND, Shukla & Marlin (2021)], and [ODE$^2$VAE, Çağatay Yıldız et al. (2019)]. We used open-source codes provided by the authors for comparison. For the Latent ODE (SDE) methods, RNN and ODE-RNN were used for the encoder structures, where the decoder structures were identically set to ODE (SDE). Table 1 shows the performance of all baseline methods compared to the proposed CSDE-TP for the reconstruction tasks. As evaluation metrics, we used mean squared errors (MSE) and negative log-likelihood (NLL) with open-source code in Rubanova et al. (2019). As shown in Table 1, the proposed method consistently outperformed the baseline methods by a large margin. In this experiment, we observed that latent dynamics-based methods (*e.g.*, Latent ODE/SDE with RNN and ODE-RNN encoders) on models attained similar performances. We set the latent dimensions of each control agent to 128 for both the reconstruction and prediction experiments. In the experiments on both datasets, the Mckean-Vlasov (MV) type of the SDE model slightly improved the performance, where it subtracted the mean (*i.e.*, mean-shifting) of the control agent outputs to normalize/reduce the intrinsic volatility in the inferred process $\hat{X}_{t_k}^{\alpha}$.

## 4.2 TIME-SERIES DATA PREDICTION

Table 2: **Evaluation of prediction tasks on the PhysioNet/Air Quality datasets.**

| Methods | PhysioNet | | Air Quality | |
| --- | --- | --- | --- | --- |
| | MSE $(\times 10^{-1})\downarrow$ | NLL $(\times 10^2)\uparrow$ | MSE $(\times 10^{-1})\downarrow$ | NLL $(\times 10^3)\uparrow$ |
| ODE$^2$VAE | 5.369 | $-26.812$ | 7.938 | $-3.965$ |
| Latent ODE (RNN-Enc) | 2.038 | $-10.153$ | 3.403 | $-1.772$ |
| Latent ODE (ODE-Enc) | 1.977 | $-9.849$ | 3.374 | $-1.683$ |
| Latent SDE (RNN-Enc) | 2.002 | $-9.975$ | 3.676 | $-1.834$ |
| Latent SDE (ODE-Enc) | 1.995 | $-9.936$ | 3.328 | $-1.660$ |
| mTAND | 1.551 | $-7.719$ | 1.867 | $-0.929$ |
| CSDE-TP | **1.195** | **$-5.937$** | **1.277** | **$-0.630$** |
| CSDE-TP-MV | **1.057** | **$-5.811$** | **1.671** | **$-0.638$** |

In this experiment, all the methods were evaluated to predict the remaining 12 time sequences (*e.g.*, 20%) on the test dataset given the first 36 time-sequences (*e.g.*, 80%), and we followed a procedure similar to that suggested by Rubanova et al. (2019) to predict the future time series. Table 2 compares the performance in terms of MSE and NLL, in which the proposed method considerably outperforms the baseline methods. In the experiment on the Air quality dataset, we set a smaller latent dimension of 64 as the feature dimension compared to those of the other datasets. Interestingly, the CSDE-TP-MV model exhibited worse performance than the vanilla CSDE-TP model when the data dimension was very small. This result indicates that the mean-shifting in a low-dimensional data space reduces the expressiveness of the CSDEs.

## 4.3 UNCERTAINTY ESTIMATION ON STOCK MARKET DATASET

When high volatility is observed over the temporal/spatial axes, conventional evaluation metrics such as MSEs hardly capture the stochastic property of the time-series variations. Thus, to capture the stochasticity, we evaluated the distance between the distributions of the test data and the inferred/generated data using the maximum mean discrepancy (MMD). We followed the protocol

Table 3: **Uncertainty estimation on the Stock Market dataset.**

| Methods | RNN | RNN-VAE | ODE-RNN | Latent ODE | Latent SDE | CSDE-TP |
|---|---|---|---|---|---|---|
| MMD $(\times 10^{-3}) \downarrow$ | 526.12 | 427.05 | 201.48 | 194.75 | 190.11 | **94.79** |

suggested by Li et al. (2017) to evaluate the MMD distance, where we used two Gaussian RBF kernels with bandwidths of $[5.0, 10.0]$. Using this evaluation metric, we experimented on reconstruction tasks using the S&P-500 Stock Market dataset. Table 3 shows that the proposed CSDE-TP outperforms baselines and effectively recovers the distributional information of stock prices with the stochastic property of the SDE models and the proposed optimization framework. Interestingly, the latent SDE model attains better performance compared to the Latent ODE, as it utilizes an additional Wiener process to model the data uncertainty. The performance improvement of the Latent SDE vanishes when we remove the diffusion term ($\sigma = 0$) of the latent SDE.

## 4.4 EMPIRICAL STUDY

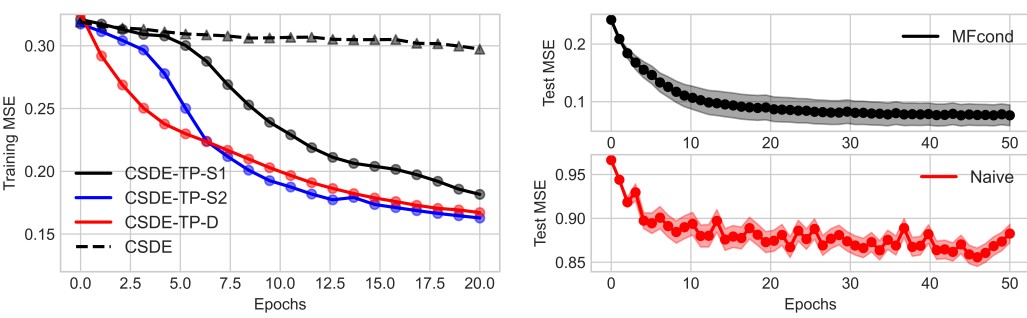

(a) Efficiency of Markov-DP with CSDE-TP      (b) Efficiency of the MFcond loss

Figure 2: **Ablation study on the effectiveness of the proposed method**.

**Efficiency of the Markov-DP-TP framework.** To show the empirical advantages of our CSDE-TP model with Markov-DP learning schemes, we evaluated our CSDE-TP according to a different number of control agents on the prediction task using the Air Quality dataset. Figure 2-(a) shows the training MSEs for several variants of the proposed model in the first 20 epochs, where CSDE-TP-**S**hallow1, -**S**hallow2, and -**D**eep (*i.e.*, black, blue, and red lines) denote the proposed models with a different number of control agents, *i.e.*, $M = 2, 8$, and $48$, respectively. The standard CSDE model (*i.e.*, the black dashed line) utilized a single agent, $M = 1$. For all models, the total number of training parameters was equivalently set to $\approx 40K$, and the number of parameters was normalized. As shown in Figure 2-(a), despite using the same number of parameters, employing multiple agents clearly outperforms the standard CSDE in terms of the learning curve. From this fact, we can conclude that the Markov-DP-TP significantly increased the network efficiency compared to the standard CSDE, which indicates that our Markov-DP framework is crucial for training controlled dynamics models.

**Efficiency of the MFcond loss.** In this experiment, we show the empirical advantages of the multi-conditioned CSDE in (8) against the naive partial-conditioned CSDE in (6). Similar to previous experiments, the results were obtained for the prediction task with the Air Quality dataset. Figure 2-(b) shows the model confidence in testing MSEs for the first 50 epochs, where shaded areas indicate the confidence regions (*i.e.*, $\pm std$). The proposed MFcond loss exhibits considerable performance improvement ($.08 \ll .87$) compared to the conventional native cost functional and reduces the variances in loss landscapes with stable learning. With the theoretical discussion in Appendix A.3, we conclude that the proposed CSDE actively exploits the information of the complex time series with multiple conditions to accurately generate complex time-series.

## 5 CONCLUSION

In this paper, we introduce a novel Markov-type CSDE with the TP function that records the individual attention of each control agent at sub-intervals along the temporal axis. Using the properties of the CSDE and TP, we suggest Markov DP to efficiently train the control agents by decomposing the original problem into smaller sub-problems. To overcome the practical/theoretical issues, we propose two novel losses, namely, MFcond and MBcond losses. The MFcond loss captures the future time to estimate the running costs, while multiple conditions are actively provided to forward dynamics. The MBcond loss assures the theoretical optimality of the control agents and imposes regularization by providing additional information to backward dynamics. Experimental results demonstrate the efficiency of the proposed method for various tasks using real datasets.

**Acknowledgments.** This work was supported by Institute of Information communications Technology Planning Evaluation (IITP) grant funded by the Korea government (MSIT) (No.2021-0-01341, Artificial Intelligence Graduate School Program (ChungAng university)).

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

## A APPENDIX

### A.1 DETAILED COMPARISON TO EXISTING METHODS

In this section, we investigate the relation between our method and existing methods.

**Reverse SDE vs. Backward SDE.** Song et al. (2020) suggested a novel SDE called *reverse SDE*, which shares semantically similar idea with BSDE: both reverse/backward SDEs enhance the forward SDE by providing additional information to drift/diffusion functions in forward dynamics.

The mathematical motivation of the reverse SDE in Anderson (1982) is to pose the SDEs with Wiener processes $W_t, \hat{W}_t$ with respect to these minimal increasing/decreasing sigma algebras $\mathcal{A}_t, \hat{\mathcal{A}}_t$ and define the relation between them:

$$d\hat{W}_t = \frac{1}{p_t(X_t)}\nabla[p_t(X_t)\sigma(t, X_t)]dt + dW_t, \tag{15}$$

where $X_t$ is a solution to the forward SDE and $p_t$ is the probability density of $X_t$. Using the relation in (15), the reverse SDE transforms the prior distribution (*e.g.*, Gaussian noise distribution) back into the data distribution (*e.g.*, 2D images) by gradually removing the noises and reconstruct the original data with the well-designed score function (*i.e.*, $\nabla p_t(x)$) in backward dynamics. In contrast to the reverse SDE, the role of backward SDE in this paper is to consider the probabilistic reformulation to access the cost functional to provide the additional information in backward dynamics.

**Stacked ODE vs. CSDE-TP.** Massaroli et al. (2020) suggested the stacked Neural ODE that shares similar idea with the proposed CSDE-TP, where temporally piece-wise neural nets are considered to model the complex dynamics. However, the stacked ODE faces the aforementioned problem on partial conditional information when generating complex data as their models only take initial values to propagate dynamics. As opposed to their models, the proposed model is trained with multiple observations and directly generates time series in data space without any latent embedding network. Furthermore, we generalize their optimal control problems to the stochastic version and propose the Markov-DP-TP framework that can systemically solve the problem.

**DDPMs vs. CSDE-TP.** Tashiro et al. (2021) suggested denoising denoising diffusion models (DDPMs) that are conditioned on the set of the observed data, where the generated sequential data is assumed to be gradually transformed from an initial state in the forward direction and the backward process is parameterized by the neural network and trained to minimize specific ELBOs.

Specifically, the transition probability $p_\theta(X_{t-1}|X_t)$ in the backward process is defined as a parameterized Gaussian distribution:

$$p_\theta(X_t|X_{t-1}) = \mathcal{N}(X_{t-1}; \mu_\theta(t, X_t), \sigma_\theta(t, X_t)), \tag{16}$$

where mean and covariance $(\mu_\theta, \sigma_\theta)$ are parameterized by the neural network $\theta$. Similar to the proposed CSDE, their parameterized functions are closed-loop type processes and the whole probabilistic sequential model $p_\theta$ is posed as the Markov chain: $p_\theta(X_{0:T}) = p_\theta(X_T)\prod_{t=1}^{T} p_\theta(X_{t-1}|X_t)$.

In contrast to the DDPM, the probability transition in the proposed CSDE is defined as the continuous generalization called *controlled Fokker-Planck equation* (CFPE):

$$\frac{\partial}{\partial t}p_\theta^\alpha(x, t|y, s) = -\nabla f(x, t, \alpha_\theta)p_t(x) + \frac{1}{2}\mathbf{Tr}\left[\nabla^2 \sigma\sigma^T(x, t, \alpha_\theta) \cdot p_t(x)\right], \tag{17}$$

where $t > s \in [0, T)$ and $x, y \in \mathbb{R}^d$, $p_t \sim X_t^\alpha$ is the probability distribution of $X_t^\alpha$. The CFPE in (17) one-to-one corresponds to the CSDE (*i.e.*, $X_t^\alpha$). Compared to the discrete-time Gaussian transition model, this conditional probability can express complex continuous-time probability transitions while maintaining the Markov structure.

### A.2 NOTATIONS AND BACKGROUND

We first define the basic definitions of probabilistic objects:

**Definition 3.** *A filtration $\{\mathcal{F}_t\}$ is an increasing sequence of $\{\mathcal{F}_t\}$ of $\sigma$-algebra such that $\mathcal{F}_0 \cdots, \subset \mathcal{F}_t \subset \mathcal{F}$. The triplet $(\Omega, \mathcal{F}_t, \mathbb{P})$ is called a filtered probability space.*

**Definition 4.** *The filtration generated by Wiener process $W_t$ is defined as $\mathcal{F}_t^W = \sigma\{W_0, \cdots, W_t\}$. In this case, $W_t$ is naturally $\mathcal{F}_t^W$-adapted by construction.*

**Definition 5.** *The stochastic process $\{X_t\}$ is called $\{\mathcal{F}_t\}$-adapted if $X_t$ is $\mathcal{F}_t$ measurable for every $0 \leq t \leq T$.*

Throughout this paper, we work on the filtered probability space $(\Omega, \{\mathcal{F}_t\}_{t \in [0,T]}, \mathbb{P})$ with the $d$-dimensional $\mathcal{F}_t$-Wiener process $W_t$ and natural filtration $\mathcal{F}_t^W$. We assume that $\alpha^i$ for all $1 \leq i \leq M$ is *admissible Markov control*, (*i.e.*, $\alpha^i$ is $\mathcal{F}_t$-adapted and $\alpha^i \in \mathbb{A}_i$, $X_t^\alpha$ has a unique solution).

**Definition 6.** *(Markov Process) Let $X_t$ be a $\mathcal{F}_t$-adapted stochastic process. Then, $X_t$ is the Markov process if the following equality holds:*

$$\mathbb{E}[X_t | \mathcal{F}_s] = \mathbb{E}[X_t | X_s], \quad \forall s \leq t. \tag{18}$$

**Definition 7.** *(Controlled Stochastic Differential Equation)*

$$X_t^\alpha = X_s + \int_s^t b\left(u, X_u^\alpha, \alpha\right) du + \int_s^t \sigma\left(u, X_u, \alpha\right) dW_u, \quad for \quad 0 \leq s \leq t \leq T. \tag{19}$$

The solution to the above CSDE is denoted as $X_t^{\alpha,s}$. If the initial states are specified (*i.e.*, starting point $X_s = x$), we denote the solution as $X_t^{\alpha,s,x}$. By the definition of Markovian control agents, in all cases, the solution to the proposed CSDE in (1) is a Markov process.

**Mathematical Assumptions.** In this paper, we assume that functions $b, \sigma$ are uniformly Lipschitz continuous along its spatial axis and bounded $\|b(t, 0; \cdot)\|, \|\sigma(t, 0; \cdot)\|$ at the entire interval $[0, T]$. We assume that each functions $b^i(\cdot, x, \cdot), \sigma^i(\cdot, x, \cdot), \Psi(x), l(\cdot, x)$ are twice differentiable for all $1 \leq i \leq M$, (*i.e.*,, $b^i, \sigma^i, \Psi, l \in C^2(\mathbb{R}^n)$. and both drift and diffusion functions are uniformly Lipschitz on its spatial axis, *i.e.*, $b^i, \partial_x b^i, \partial_x^2 b^i, \sigma^i, \partial_x \sigma^i, \partial_x^2 \sigma^i \in \mathbf{Lip}$), and the trainable parameters of the control agents $\alpha^i, \theta^i$ are lying in the compact subset $\mathbb{C}$ of its ambient space (*i.e.*, $\theta^i \in \mathbb{C} \subset \mathbb{R}^m$).

### A.3 Enlarged Information by Collection of Observed Data

In the proposed inference procedure, we define a novel operator $\mathcal{T}$ in (9) to consider the multi-conditioned dynamics with the Markov-type SDE model. Although this operator plays a central role in the paper, its mathematical properties have not been carefully dealt and investigated thoroughly. In this section, we discuss the relation between this operator and the *enlarged information* that is obtained by collecting past observations. In addition, we generalize the inference mechanism in (9) to a mathematically rigorous form and discuss the effect of the proposed operator $\mathcal{T}$ by showing some probability inequality.

Suppose that we have two observed conditional states $\{X_{t_m}\}, \{X_{t_n}\}$ until the current time $t$, $(t_n, t_m < t < t_k)$ and the objective is to predict/generate the future value $y_{t_k}$ using this information. We consider the deterministic time $t_k$ by replacing random stopping time $\tau_{t_m}$ to simplify the discussion. First, we define the two-parameter stochastic process $Y$ to model the proposed operator $\mathcal{T}$ in an alternative way:

$$\mathcal{T}_{t_m, t_k} = Y(t_m, t_n)(w) \triangleq \frac{1}{2}\left(X_{1, t_k}^{\alpha, t_m}(w) + X_{2, t_k}^{\alpha, t_n}(w)\right), \tag{20}$$

where $w \in \Omega$ takes a value in the probability space. The stochastic process $Y$ is the $(\mathcal{F}_{t_m} \vee \mathcal{F}_{t_n})$-valued random variable for any fixed $t_m, t_n < t$ by the definition, where $\mathcal{F}_{t_m} \vee \mathcal{F}_{t_n} \triangleq \Sigma(\mathcal{F}^{\mathbb{M}_1} \cup \mathcal{F}^{\mathbb{M}_2})$ is the composited smallest sigma algebra with two filtrations. In the definition, we assume that processes $X_{1, t_k}^{\alpha, t_m}, X_{2, t_k}^{\alpha, t_n}$ are derived from two independent Wiener processes $W_t$ and $\hat{W}_t$. Then, we can define the *two-parameter martingale* Zakai (1981); Khoshnevisan (2003) in the following form:

$$\mathbb{M}(t_m, t_n)(w) = \mathbb{E}\left[l(t_k, Y(t_m, t_n)) | \mathcal{F}_{t_m} \vee \mathcal{F}_{t_n}\right]. \tag{21}$$

By the definition of $\mathbb{M}$, it can be easily shown that $\mathbb{M}$ is the reformulation of the MFcond loss for some fixed number of past observations. Note that $\mathbb{M}$ is truly a martingale because conditional estimations are summed in the definition of $\mathcal{T}$. The control agents are trained to minimize the $\mathbb{M}$ given the information induced by past observations (*i.e.*, composited filtration $\{\bigvee_{t_m < t} \mathcal{F}^{\mathbb{M}_{t_m}}\}$), which indicates that the proposed inference procedure can infer the future value $\hat{X}_{t_k}^\alpha$ according to the enlarged information $\{\bigvee_{t_m < t} \mathcal{F}^{\mathbb{M}_{t_m}}\}$. By the fact that $\mathbb{M}$ is a martingale with respect to the composited filtration, we obtain the following result using Doob's maximal inequality:

$$1 - \frac{1}{2\eta}\left(\mathbb{E}\left[\|X_{1, t_k}^{\alpha, t} - y_{t_k}\|\right] + \mathbb{E}\left[\|X_{2, t_k}^{\alpha, t} - y_{t_k}\|\right]\right) \leq \mathbb{P}\left[\sup_{t_n < t}\sup_{t_m < t}\mathbb{E}[l \circ \mathcal{T} | \mathcal{F}_{t_m} \vee \mathcal{F}_{t_n}] \leq \eta\right], \tag{22}$$

where the inequality shows that errors between the future value $y_{t_k}$ and the generated samples $X_{t_k}^{\alpha,t}$ at time $t_k$ are bounded by the maximal perturbation probability. As the control agents are trained to minimize the MFcond loss (*i.e.*, $\mathbb{M}$) in the right-hand side of inequality, it renders the probabilistic bound of $L_2$ errors at future time $t_k$.

## A.4 DETAILED DISCUSSIONS ON THE MBCOND LOSS

In this section, we investigate the detailed theoretical structure of the MBcond loss and its fundamental rationale for the optimality of control agents. For this, we rephrase the cost functional in the general form:

$$J(t,x) = \mathbb{E}\left[\int_t^T l(s, X_s^\alpha)ds + \Psi(X_T^\alpha)\Big| X_t = x\right]. \tag{23}$$

The classical non-linear Feynman-Kac theorem in Yong & Zhou (1999) states that given the cost functional $J$ with the control agents $\alpha$, one can obtain the second-order parabolic partial differential equation from (23):

$$\frac{\partial J}{\partial t} + \langle \nabla J, b(t,x,\alpha)\rangle + \frac{1}{2}\mathbf{Tr}\left[\sigma\sigma^T(t,x,\alpha)\nabla^2 J\right] + l(s,x) = 0, \tag{24}$$

where $\langle\cdot,\cdot\rangle$ denotes the inner product and the boundary condition is given as $J(T,x) = \Psi(x)$. Subsequently, by applying Itô's formula to (23), we obtain the following probabilistic formulation:

$$\Psi(X_T^\alpha) = J(s, X_s) + \int_t^T \left[\frac{\partial J}{\partial t}(s, X_s) + \frac{1}{2}\mathbf{Tr}[\sigma\sigma^T(s, X_s, \alpha)\nabla^2 J] + \langle b(s, X_s, \alpha), \nabla J\rangle\right] dt$$

$$+\int_t^T \langle \sigma^T(s, X_s, \alpha)\nabla J(s, X_s), dW_t\rangle = -\int_t^T l(s, X_s)ds + \int_s^T \langle\sigma^T(s, X_s, \alpha)\nabla J(s, X_s), dW_s\rangle. \tag{25}$$

By rearranging each term above, the backward stochastic differential equation is induced directly.

$$Z_t^\alpha = J(t, X_t^\alpha) = \Psi(X_T^\alpha) + \int_t^T l(s, X_s)ds - \int_t^T \langle\sigma^T(t, X_t, \alpha)\nabla J(s, X_s), dW_s\rangle. \tag{26}$$

Note that, in the main paper, we use the inverse sign convention, $\int_t^T(\cdot) = -\int_T^t(\cdot)$, to emphasize the backward direction. By using the formulations, the MBcond loss in (13) can be rewritten in full description as follows:

$$\mathcal{L}_b(\alpha, Z_t^\alpha) = \int_{[0,T]} \mathbb{E}_{\mathbf{y}(\cdot)}\left[\left|\Psi(X_T^\alpha) + \int_t^T l(s, X_s^\alpha)ds - \int_t^T \langle\sigma^T(t, X_t^\alpha, \alpha)\nabla J(s, X_s^\alpha), dW_s\rangle\right|^2 \Big| X_t^\alpha = y_t\right]dt, \tag{27}$$

where $\mathbf{y}_{(\cdot)} = (y_1, \cdots, y_T) \sim p(y_1, \cdots, y_T)$ denotes the set of observed data. The regularization effect comes from the expectation evaluation of the third term in (26). Specifically, one can obtain the following equality by using the Itô's isometry:

$$\mathbb{E}\left[\left|\int_t^T \langle\sigma^T(t, X_t, \alpha)\nabla J(X_s), dW_s\rangle\right|^2 |X_t\right] = \mathbb{E}\left[\int_t^T \left\|\sigma^T(t, X_t, \alpha)\nabla J(s, X_s)\right\|^2 dt|X_t\right]. \tag{28}$$

Because the MBcond loss is posed to minimize this additional martingale term in (28) in backward dynamics according to the forward dynamics $X_t^\alpha$, it reduces the over-confidence of generated time-series. By the relation $(t,x) \to J(t,x) \to Z_t^\alpha \to \mathcal{L}_b(t,x)$ for any $(t,x) \in [0,T] \times \mathbb{R}^+$, the update rule for the MBcond loss can be expressed as follows:

$$\theta_{k+1}^r = \theta_k^r - \frac{\partial}{\partial\theta^r}\left[\mathcal{L}_b\left(s, X_s^{\alpha^r(\cdot,\cdot,\theta_k^r)}\right)ds\right], \tag{29}$$

where this formulation is similar to (5) and shows that gradient descent with respect to $\theta^r$ for the MBcond loss can be explicitly defined.

**Admissible Control Set** $\mathbb{A}$. In previous discussions, we show the relation between $J$ with BSDE dynamics $Z_t$ and the well-defined gradient descent. The next step is to define the proper control set $\mathbb{A}$ to relate the gradient descent with optimality.

Let us define the Hilbert space $\mathcal{L}^2 \triangleq \{\varphi(t, x; \bar{\theta}); \mathbb{R}^n\text{-valued } \mathcal{F}_t\text{-progressively measurable } \forall \bar{\theta} \in \mathbb{C}\}$ with the norm $\|\varphi\|_{\mathcal{L}^2}^2 = \mathbb{E}\left[\int_0^T |\varphi(t, x; \theta)|^2 dt\right] < \infty$. We assume that control agents $\alpha^r$ is $L_r^\alpha$ Lipschitz on the parameter variable, $i.e.,$ $\left\|\alpha^r(\cdot, \cdot; \theta_{k,1}^r) - \alpha^r(\cdot, \cdot; \theta_{k,2}^r)\right\|_{\mathcal{L}^2} \le L_r^\alpha \left\|\theta_{k,1}^r - \theta_{k,2}^r\right\|$ for any $\theta_{k,1}^r \ne \theta_{k,2}^r \in \mathbb{R}^m$ and any $1 \le k \le K$. In all case, we assume that any $\theta_k^r$ lies in the compact subset $\mathbb{C}$ of $\mathbb{R}^m$. Each functions $b^i(\cdot, x, \cdot), \sigma^i(\cdot, x, \cdot), \Psi(x), l(\cdot, x)$ are twice differentiable for all $1 \le i \le M$, $(i.e., b^i, \sigma^i, \Psi, l \in C^2(\mathbb{R}^n)$. and both drift and diffusion functions are uniformly Lipschitz on its spatial axis, $i.e., b^i, \partial_x b^i, \partial_x^2 b^i, \sigma^i, \partial_x \sigma^i, \partial_x^2 \sigma^i \in \mathbf{Lip}$). As we defined $\Psi$ and $l$ as usual Euclidean distance, regularity/uniform Lipschitzness for these functions are trivial.

For the fixed parameter $\theta$, we define the $r$-th control agent as $\theta^r \to \alpha^r(\cdot, \cdot, \theta^r) \triangleq \alpha^r(\theta) \in \mathcal{L}^2$. Truly, the image space of $\alpha^r(\theta)$ is the closed subspace of the Hilbert space $\mathcal{L}^2$ due to the Lipschitzness with compactness of $\theta$.

Let $\theta_r(k) : \mathbb{N} \to \mathbb{R}^m$ be the trajectory for the training parameters of the $r$-th control agent at learning iteration $k$. Without loss of generality, $J_r[\alpha_r] = J(t, X_t^{(\alpha^r, \alpha^{(-r)})})$. We define the Euclidean closed metric balls $\{B_{\delta_r^k}\}_{k \in \mathbb{N}}$ centered at $\theta(k)$ with the radius $\delta_r^k < \infty$ such that $B_{\delta_r^k} = \{\vartheta \in \mathbb{R}^m; \|\vartheta - \theta_r(k)\| \le \delta_r^k, \theta_r(k) \text{ is local minimum of } J_r[\theta(k)]\}$. Let us consider the sub-sequence $\{\theta(\bar{k})\}_{\bar{k} \in \bar{N}} \subseteq \{\theta_r(k)\}_{k \in \mathbb{N}}$, which induces the strictly-decreasing cost functional $\{J_r[\theta(\bar{k})]\}_{\bar{k} \in \bar{N}}$ with the ordered index set $\bar{N}$. Then, the admissible control set $\mathbb{A}$ is defined as follows:

$$\alpha \triangleq [\alpha^1, \cdots, \alpha^r \cdots, \alpha^M] \in \mathbb{A} = \bigotimes_{r=1}^M \left\{ \bigcap_{K}^{K} \bigcup_{\bar{j}=1}^{\bar{K}} \alpha^r(\cdot, \cdot, B_{\delta_{\bar{j}}}); \bar{j} \le \bar{K} \in \bar{N} \right\} \subset \bigotimes_{r=1}^M \mathcal{L}^2, \quad (30)$$

where $\bar{K}$ is the maximal element in $\bar{N}$ and the constant $K \in N$ indicates the last iteration index of training defined in Algorithm 1. Intuitively, the control set $\mathbb{A}$ can be understood as a collection of local minimum obtained by $M$ gradient descent schemes during training.

$$V \triangleq J[\alpha_\star] = J[\alpha(\theta(K))] = \inf_{\alpha \in \mathbb{A}} J[\alpha(\theta)]. \quad (31)$$

where $V \in C^{1,2}([0, T], \mathbb{R}^d)$. By the definition of metric balls $\{B_{\delta_k}\}$ and strictly-decreasing properties, the infimum in (31) is attained when $\theta(K) = \theta$ and the control agent $\alpha(\theta(K))$ is optimal in this control set.

**Relation to Stochastic Maximum Principle (SMP).** We consider an arbitrary control in the convex set $\mathbb{K} \in \mathbb{A}$ with $\beta \in \mathbb{K}$ and the optimal control $\alpha(\theta(K))$. Let $DJ|_\beta = \frac{d}{d\epsilon} J(\theta(K) + \epsilon(\beta - \alpha))|_{\epsilon=0}$ be the Gâteaux derivative (this can be defined while the control set is a vector sub-space, $\mathbb{A} \subset \mathcal{L}^2$). By the result of Pontryagin maximum principle, Theorem (4.12) in Carmona (2016a), one can obtain the inequality as follows:

$$\left\|\frac{\partial}{\partial \alpha} \mathcal{H}\right\| \cdot (\vee_r^M L_r^\alpha) \|\theta(K) - \theta\| \ge \left\|\frac{\partial}{\partial \alpha} \mathcal{H}\right\| \cdot \|[\alpha(t, X_t, \theta(K)) - \beta(t, X_t, \theta)]\| \ge DJ|_\beta \ge 0 \quad (32)$$

for $t \in [0, T]$ almost surely, where we define $\mathcal{H} \triangleq \mathcal{H}(t, X_t, Y_t, Z_t, \alpha_t)$ for the Hamiltonian system $\mathcal{H}$ with adjoint variables $Y_t, Z_t$ and define the arbitrary control $\beta = \alpha(\cdot, \cdot, \theta) \in \mathbb{A}$ with some $\theta$. The first inequality is satisfied due to the definition of Lipschitzian control agents. The optimality condition indicates converging upper-bound of $DJ|_\beta$ to 0.

In our method, the optimality condition of the proposed learning framework is bounded by the Euclidean distance between $\theta(K)$ and $\theta$ in parameter space. Thus, the proposed framework poses a fundamentally different approach to interact with optimality conditions in SMP. As we define that the $\theta(K)$ is a local minimum of $J$ with the inequality $\|\theta(K) - \theta\| < \delta_\theta^K$, the gradient descent scheme that induces the tight radius $\{\delta_\theta^k\}_{k \in \mathbb{N}^+}$ assures optimality by the relation $0 \le DJ|_\beta \approx \delta_\theta^K$.

**Relation to the HJB equation.** We consider the infinitesimal generator $\mathcal{L}_t$ of the non-homogeneous controlled Markov process $X_t$ as $\mathcal{L}_t^\alpha f = \langle \nabla f, b(t, x, \alpha) \rangle + \frac{1}{2} \mathbf{Tr}\left[\sigma \sigma^T(t, x, \alpha) \nabla^2 f\right]$. We show the important relation between the proposed MBcond loss with the HJB equation as follows:

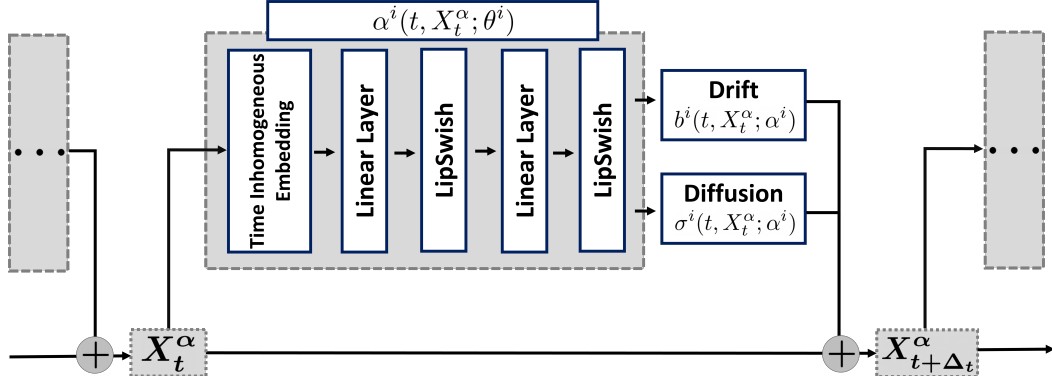

Figure 3: **The pipeline of the proposed CSDE-TP with neural control agents**

**Equivalence Relation**

$$\underbrace{\frac{\partial J}{\partial t}(t,x) + \mathcal{L}_t^{\alpha(\theta(K))} J(t,x) + l(t,x) = 0}_{\text{Non-linear Feynman-Kac, MBcond loss}} = \underbrace{\frac{\partial V}{\partial t}(t,x) + \inf_{\alpha \in \mathbb{A}} [\mathcal{L}_t^{\alpha} V(t,x) + l(t,x)]}_{\text{HJB equation, exact solution}} \quad (33)$$

In the left-hand side of (33), the PDE formula is directly consequence of the non-linear Feynman-Kac theorem that we derive in (24). The distinct point is that control agents are obtained by the gradient descent of the MBcond loss with BSDE (*i.e.*, $Z_t$). Note that, as shown in (31), $\theta(K)$ is actually an optimal control. This means that, without heavy calculations to solve the PDEs, the gradient descent algorithm also assures optimality of control agents in the proposed control set $\mathbb{A}$.

In contrast, the HJB equation in the right-hand side states that the optimal control agent can be obtained by solving the second-order parabolic formula and the infimum is taken by considering algebraic properties of candidates for the exact solution. If the solution to HJBE exists in the control set $\mathbb{A}$, the PDE in the left hand side of (33) approximates the solution to the HJB. Overall, we argue that the MFBcond loss can provide a novel deep learning-based paradigm to adopt/solve the conventional stochastic optimal control problem in a feasible way (*i.e.*, well-defined loss functions with the gradient descent scheme).

### A.5 Data Prepossessing

*PhysioNet dataset*, Silva et al. (2012), contains overall 8000 multivariate time series obtained for the first 48 hours of a different patient's admission to intensive care unit (ICU). Each patient has a set of 35 various clinical features. We normalized features of all patients in the dataset to have zero mean and unit variance. We used a half of time-series as the training dataset and the remaining parts as the test dataset.

*Speech Commands dataset*, Warden (2018), consists of one-second audio records of various spoken words such as "Yes", "No", "Up", and "Down". Since there were nearly 100,000 record samples, we sub-sampled the dataset due to the dimensionality of training instances on two conflicting classes (*i.e.*, "Right" and "Left"). Overall, 6950 time-series records were selected, where $80\%$ were used as training dataset and the remaining parts as the test dataset. We pre-processed these time series by computing Mel-frequency cepstrum coefficients from the audio signal, so that each time series was spaced with 65 and 54 channels. Then, we normalized each channel of all signals in the dataset to have zero mean and unit variance.

*Beijing Air-Quality dataset*, Zhang et al. (2017), consists of multi-year recordings of air quality data across different locations in Beijing. Each sample contains 6-dimensional time series features of $PM_{2.5}$, $PM_{10}$, $SO_2$, $NO_2$, $CO$, and $O_3$, which are recorded per hour. We segmented data to have the length of 48 and normalized each feature of all data in the dataset to have zero mean and unit variance.

*S&P-500 Stock Market dataset* consists of stock market data with 6-dimensional feature vectors (*i.e.*, [High, Low, Open, Close, Volume, Adj Close]). For the complete data acquisitions, we excluded enterprises with incomplete recordings during sampling duration, thus total 381 enterprises are selected. The time-series are sampled every 30-min with $\mathbb{T} = 48$ temporal states. Similar to Speech commands dataset, we used first $80\%$ of temporal states to train the model and the remaining parts are used for prediction task.

## A.6   EXPERIMENTS DETAILS

**Different SDE candidates for CSDE.** Owing to the abstract form of the proposed CSDE in (1), various types of drift and diffusion functions (*i.e.*, $b$ and $\sigma$) can be selected according to different applications. In Table 4, we enumerate candidate functions. In the experiments, we adopted two models: Vanilla and Mckean-Vlasov (MV) SDEs.

**Hyperparameters.**  For the running and terminal costs ($l$ and $\Psi$, respectively), we used the $l_2$ distance, *i.e.*, $l(s, x) = \|x - y_s\|_2^2$ and $\Psi(x) = \|x - y_T\|^2$. In all experiments, $\gamma$ is set to $0.95$. To estimate the gradient of the MBcond loss, we estimated numerical gradients with the auto-grad library in `Pytorch` (Paszke et al. (2019)).

**Network Architecture for Neural Control Agents.**
Each control agent $\alpha^i(t, X_t; \theta^i)$ has an identical neural network architecture, which consists of linear layers and non-linear units. Figure 3 shows the detailed network architecture. Each agent takes concatenation of temporal/spatial tensors $(t, X_t)$ as its input, where the temporal tensor $t$ is transformed into new form $t'$ by the time inhomogeneous embedding layer. We followed the setting suggested in Park et al. (2021) for this embedding. After time embedding, concatenated tensor $(t', X_t)$ is fed into two Linear layers with non-

Table 4: **Examples of Neural CSDEs**.

| SDE Type | $b^i, \sigma^i$ |
|---|---|
| Vanilla | $\alpha^i(t, X_t^\alpha)$ |
| Langevin | $\nabla \alpha^i(X_t^\alpha), \sqrt{2}\sigma^i$ |
| Ornstein-Ulenbeck | $[\mu^i - \alpha^i(X_t^\alpha)], \sigma^i$ |
| McKean-Vlasov | $[\mathbb{E}[\alpha^i(t, X_t^\alpha)] - \alpha^i], \sigma^i$ |

linearity units (*i.e.*, LipSwish in Chen et al. (2019); Kidger et al. (2021)). Finally, the transformed tensors are split into the control terms for drift and diffusion functions. The diffusion functions are defined as non-degenerate types, where $\sigma^i(t, X_t, \alpha^i) = \mathbf{Diag}(z_t)$ and $z_t$ is the output of the last linear layer. The latent dimension of each Linear layer was set to 128 in all experiments except for the prediction task with the Air Quality dataset ($= 64$). Thus, a total number of training parameters for single control agent $\alpha^i$ is $\approx 11K$.

**Simulation of CSDE and Temporal Privacy Function.** Let $\mathbb{T} = t \in \{t_k\}_{1 \leq k \leq N}$ for the pre-fixed time interval $\Delta_t$. We apply the Euler-Maruyama scheme to approximately simulate the proposed CSDE:

$$X_{t+\Delta_t}^\alpha = X_t^\alpha + \sum_{i=1}^M w^i(t) b^i(t, X_t^\alpha, \alpha^i(t, X_t^\alpha; \theta^i))\Delta_t + \sum_{i=1}^M w^i(t)\sigma^i(t, X_t^\alpha, \alpha^i(t, X_t^\alpha; \theta^i))\mathbf{Z}, \quad (34)$$

where $\mathbf{Z} \triangleq \mathbf{Z}(0, \sqrt{\Delta_t}I_d)$ is a $d$-dimensional Gaussian random variable with zero mean and covariance $\sqrt{\Delta_t}I_d$.

**Analysis of Instability at Contact Points.** In every time stamps $t$, drift and diffusion functions are controlled over neural control agents $\alpha^i$ where we assume that $t_-$ and $t_+$ are adjacent points of contact point $t$ with infinitesimally small duration. The process $A_t$ indicates drift integral term and $\sigma_s^\alpha$ denotes the diffusion term in our forward CSDE dynamics. As shown in the inequality, the Markovian property is still preserved, and the magnitude of jumps are controlled by Lipschitzness of drift/diffusion functions.

$$\mathbb{E}\left[\left\|X_{t_-} - X_{t_+}\right\|^2 | \mathcal{F}_{t_-}\right] \leq \mathbb{E}[\|A_t\|^2 | X_{t_-}] + \mathbb{E}\left[\int_{t_-}^t \|\sigma_s^\alpha\|^2 ds | X_{t_-}\right] + \mathbb{E}\left[\int_t^{t_+} \left\|\sigma_s^\beta\right\|^2 ds | X_{t_-}\right].$$
$$(35)$$

In a probabilistic point of view, the set of contact points may be regarded as measure-zero, and the probabilistic evaluation is not changed.

Figure 4 shows a particular example, where 14 temporal states (*i.e.*, $|\mathbb{T}| = 14$) with 4 control agents are considered. In the figure, the black line indicates the trajectory of time series, blue dots denote the observed data points, and shaded grey dots denote the missing data points. Each control agent takes 5 data points, where 2 temporal states are shared to other agents. In the experiments, the total number of temporal privacy functions are maximally set to $M = |\mathbb{T}|/2$, where each control agent shares 2 points for smooth transitions of stochastic dynamics.

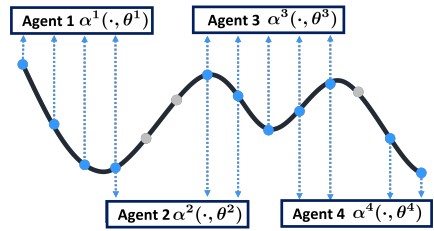

Figure 4: **The example of the temporal privacy function.**

## A.7 ADDITIONAL EMPIRICAL STUDY

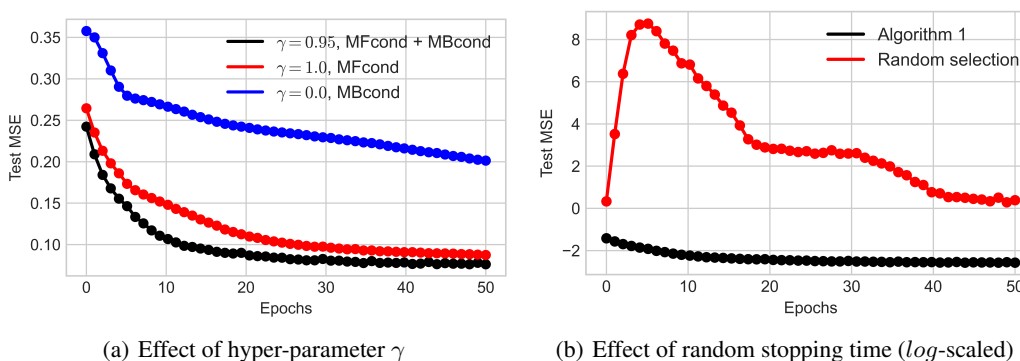

(a) Effect of hyper-parameter $\gamma$

(b) Effect of random stopping time ($log$-scaled)

Figure 5: **Ablation study on the effectiveness of the proposed method**.

**Effect of hyper-parameter $\gamma$.** In Figure 5-(a), the effect of hyper-parameter $\gamma$ is shown. Similar to Fig 2-(b), the results were obtained for the prediction task with the Air Quality dataset. Each red, black, and blue line indicates the test MSE for different $\gamma \in [0.0, 0.95, 1.0]$ over 50 epochs. If the MFcond is deactivated during the training time *i.e.*, $\gamma = 0.0$, only MBcond loss is utilized to train the proposed CSDE-TP, and the model produces poor results. As our inference procedure requires the model to train with multiple conditions, the obtained result seems obvious. If the MBcond loss is deactivated during the training time *i.e.*, $\gamma = 1.0$, multi-conditioned information in backward dynamics $Z_t^{\alpha}$ are canceled, and the performance is decreased significantly *i.e.*, $1.277 \rightarrow 2.003$. This clearly shows that MBcond loss boost the performance.

**Effect of random stopping time.** In Figure 5-(b), the effect of strategy to select $\epsilon$ is shown. If we select threshold $\epsilon$ as an uniform random variable $\epsilon \sim U[s, T]$ which is independent to $X_t$, then the network quickly falls into instability as shown in the red line of Figure 5-(b). This shows that the well-designed strategy for selecting threshold $\epsilon$ is crucial factor to stabilize the network learning landscape. Contrary to the random sampling strategy, our method defined in Algorithm 1 select half value of maximal MFcond loss in last learning steps as the threshold $\epsilon$ for random stopping time (*i.e.*, $\frac{1}{2} \max l_{k-1} \rightarrow \epsilon_k$). As the threshold $\epsilon$ is always bounded above the maximal loss in the last steps, random stopping time at iteration $k$ is decided in the time set:

$$\tau_s^k \in \{t : l(t, \mathcal{T}_{s,t}^{\alpha_k}) > \frac{1}{2} \max l(t, \mathcal{T}_{s,t}^{\alpha_{k-1}})\}, \tag{36}$$

where $\tau_s^k$ denotes the stopping time at learning iteration $k$. If the network trains the MFcond loss so that $l_k \triangleq l(t, \mathcal{T}_{s,t}^{\alpha_k}) \rightarrow 0$ as training proceeds $k \rightarrow \infty$, then it is clear that the stopping time vanishes $\tau_s^{k \rightarrow \infty} \in \emptyset$. Thus, the strategy in (36) is well-defined.

## A.8 DETAILED EXPLANATIONS OF MARKOV DYNAMIC PROGRAMMING WITH TEMPORAL PRIVACY

For the clear explanation of proposed Markov-DP-TP, let us consider the detailed example. we decompose sub-problem ($\mathbf{B}'$) in (4) into another smaller sub-problems:

$$\underbrace{\inf_{\alpha^{(-r)}} \mathbb{E}[J(u, X_u^\alpha)]}_{(\mathbf{B}')} = \underbrace{\inf_\beta \mathbb{E}\left[\int_u^{u'} l(s, X_s^\alpha)ds\right]}_{(\mathbf{C})} + \underbrace{\inf_{\beta^{(-r')}} \mathbb{E}\left[J(u', X_{u'}^\alpha)\right]}_{(\mathbf{C}')}, \tag{37}$$

where we set $\alpha^{(-r)} = \beta$, $w_r(s) = \mathbf{1}_{t \le s \le u}$. In this case, the problem $(\mathbf{B})$ on interval $[u, T]$ is now decomposed into smaller sub-problems $(\mathbf{C}), (\mathbf{C}')$ on two intervals $[u, u']$ and $[u', T]$. Similarly to $u$ in (4), another auxiliary time index $u'$ is considered here for additional problem $(C)$. The corresponding new temporal privacy function $w_{r'}(s) = \mathbf{1}_{u \le s \le u'}$ is defined on the interval $[u, u']$.

By repeating temporal decomposition of original problem $(\mathbf{A})$ $M$ times, one can find the following hierarchical relations:

- **P1**). original problem, $\mathbb{T} = [t, \cdots, T]$, $\alpha$, no temporal privacy
- **P2**). Two sub-problems $(B) + (B')$ in (4),
  Time set, $\mathbb{T} = [t, \cdots, u, \cdots, T]$,
  control agents, $\alpha = [\alpha^r, \alpha^{(-r)}]$,
  temporal privacy functions = $\{w_r\}$
- **P3**). Three sub-problems $(B) + (C) + (C')$,
  Time set, $\mathbb{T} = [t, \cdots, u, \cdots, u', \cdots, T]$,
  control agents, $\alpha = [\alpha^r, \beta, \beta^{(-r')}]$,
  temporal privacy functions = $\{w_r, w_{r'}\}$
- **P4**). $M$ sub-problems, $(A) + (B) + (C) + \ldots$,
  Time set, $\mathbb{T} = [t, \cdots, \frac{T-t}{M}, \cdots, r * \frac{T-t}{M}, \cdots, T]$,
  $\alpha = [\alpha^1, \alpha^2, \cdots, \alpha^r, \cdots, \alpha^M]$,
  temporal privacy functions = $\{w_1, w_2, \cdots, w_r, \cdots, w_M\}$

The role of $u$ in (3) and (4) is replaced to $u$ and $u'$ in $(\mathbf{P3})$, and replaced to $(r * \frac{T-t}{M})$ in $(\mathbf{P4})$ in the table if the time interval is assumed to be regularly sampled. Similarly, the role of $r$ in (3) and (4) is replaced to $r$ and $r'$ in $(\mathbf{P3})$.

## A.9 Toy Example on Synthetic Data

In this section, we conduct the reconstruction experiment on synthetic data to show the different behaviors and demonstrate the advantages of the proposed CSDE compared to previous methods.

**Stochastic Trigonometric Data.** In this experiment, we define the 100-dimensional stochastic process with composition of trigonometric functions (*i.e.*, $\sin, \cos$) as follows:

$$Y_t = \left[\frac{1}{2}\sin(5\pi t + Z_1 t) + 0.25\cos\left(\frac{13}{5}\pi t + Z_2 t\right) + Z_3\right] \in \mathbb{R}^{100}, \tag{38}$$

where we assume $t \in [0, 1.0]$ and the total number of temporal states are set to 48 (*i.e.*, $|\mathbb{T}| = 48$). In the definition of synthetic process $Y_t$, both the period and amplitude are randomized with mean-zero Gaussian random variables (*i.e.*, $Z_1 \sim \mathcal{N}(0, 1.0), Z_2 \sim \mathcal{N}(0, 2.0), Z_3 \sim \mathcal{N}(0, \frac{1}{2}I_d)$). With the effect of Gaussian random variables, the process contains high volatility in both the spatial/temporal axes. We compare our method to the auto-regressive ODE-RNN Rubanova et al. (2019) model using the open-source code implemented by the authors. To observe the fundamental difference between ODE-RNN and CSDE-TP, we stop the training procedure when the estimated MSEs of both models attained the threshold ($\le .07$). In Figures 6 and 7, the first axis of trigonometric data are visualized. The results of each model are indicated as the blue lines (*i.e.*, $X_t$) and the synthetic trigonometric data is indicated as the red lines (*i.e.*, $Y_t$). The 95%-confidence regions (*i.e.*, CR-95) of both the test and predicted time-series are shown as red and blue shaded regions, respectively.

**ODE-RNN.** Figure 6 shows the results of the ODE-RNN model. Although the ODE-RNN model attains relatively similar MSEs compared to the proposed model, there are two main issues in their model to be discussed.

1) It hardly captures the vertical perturbation of test data induced by $Z_3$ and the obtained result produces a small variance at every temporal states.

2) It hardly captures the horizontal perturbation of test data induced by $Z_1, Z_2$, and the obtained result produces the temporally unmatched trajectories.

These phenomenons occurred due to the deterministic property of the ODE-RNN model, where the dynamical transition in their model is posed as the ODEs that cannot express the stochastic variation.

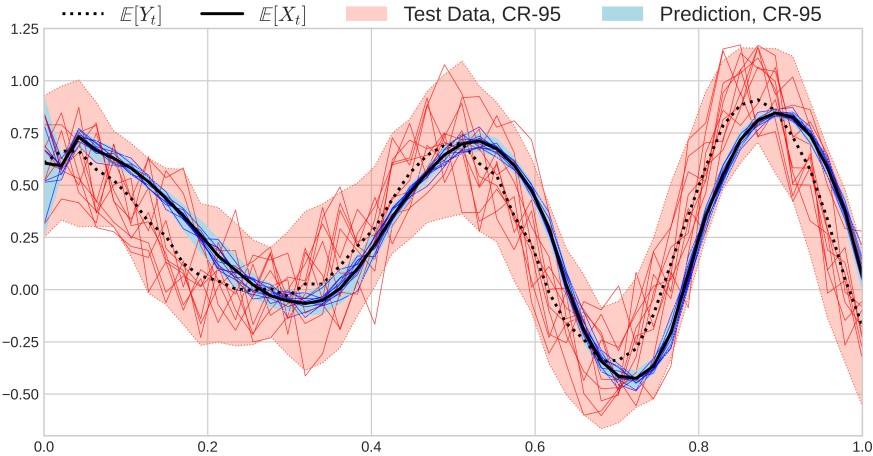

Figure 6: **ODE-RNN**.

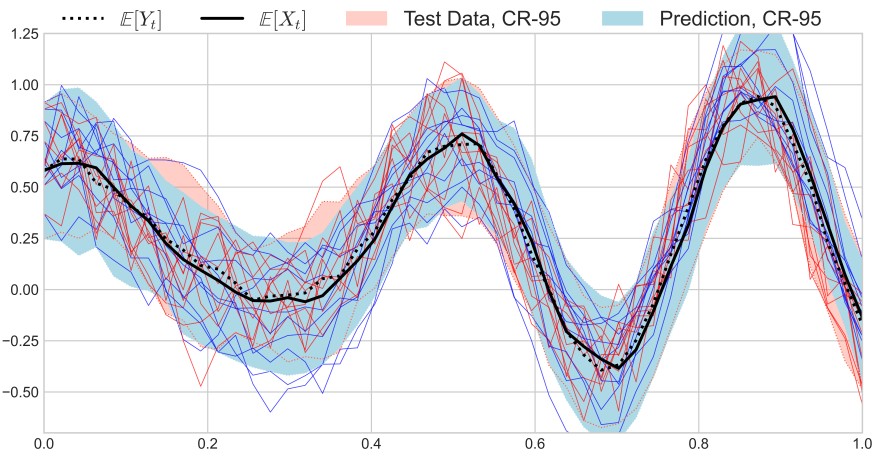

Figure 7: **CSDE-TP**.

**CSDE-TP.** Figure 7 shows the result of the proposed CSDE-TP model and shows the advantages of adopting the SDE in modelling stochastic dynamics. Compared to the results of the ODE-RNN, the proposed method accurately captures both the vertical/horizontal perturbations and recover the $95\%$ confidence region. It is clear that our CSDE-TP delicately expresses the complex volatility of stochastic trajectories.

**Discussions.** As aforementioned in Section 4.3, experimental results on synthetic stochastic data show that the MSE is not the best metric to train/evaluate the time-series models if there exists the

high volatility in the dataset. In this case, distributional metrics such as MMD and Wasserstein distance can be good substitutes for training/evaluating stochastic data.

## A.10    FUTURE WORK

We plan to extend the proposed CSDE model to a general controlled Markov Itô-Lévy jump diffusion model (Øksendal & Sulem (2007)) to delicately express the complex time-series data. For example, the proposed CSDE can be generalized to the Markov Itô-Lévy jump diffusion of the following form:

$$dX_t^\alpha = b(t, X_t^\alpha, \alpha(\theta))dt + \sigma(t, X_t^\alpha, \alpha(\theta))dW_t + \int \Gamma(t, Z)\mathbb{N}(dt, dZ), \tag{39}$$

where $\mathbb{N}(t, z) = \sum_{0 < s \leq t} \mathcal{X}_{z \in U}(\eta_s - \eta_{s^-})$, and Poisson random measure $\eta_t$. As the previous work in Jia & Benson (2019) show the effectiveness of the jump process in modelling complex discontinuous dynamics, we believe this generalization will produce the comparable results and broaden our understanding in modelling dynamical systems for time-series data.

