# OpenReview forum: "Neural Markov Controlled SDE: Stochastic Optimization for Continuous-Time Data"
_ICLR.cc/2022/Conference — ICLR 2022 Poster_

### Official Review · Reviewer_WGcS · 2021-10-30

**Correctness:** 3
**Technical Novelty And Significance:** 4
**Empirical Novelty And Significance:** 2
**Recommendation:** 8
**Confidence:** 3

**Main Review:**

The paper looks pretty solid and novel in the theoretical part. The MBcond loss built on the forward-backward stochastic differential equations and implemented using deep neural nets are new to the area of time series analysis/prediction. The authors also explain/analyze connections between the proposed model and the HJB equation and the non-linear Feynman-Kac theorem.



Major issues:

1. Motivation of temporal privacy is not clear. For each agent, it seems that it will only be valid for a specific time interval. The authors may want to explain when a control agent will kick in and when it will be invalid. At the time a control agent kicks in, will it cause jump of the stochastic process and is it still a Markov process in this case?

In addition, the 'temporal privacy' constrains the agents cannot simultaneously control the dynamics, which requires the equations (1) be a specific controlled stochastic differential equation. Therefore, the multiple control agents can be considered as one single agent.

2. Eq. (4), why the expected terminal loss J given X_t^{\alpha} is optimized using \alpha^{-r}? Due to causality,  I think it should only depend on partial control agents which are valid after time u.

3. Page 4, Definition 1. For each observation y, the dependent forward loss is calculated till time t as shown in the integration. So observations sampled at the time far away from t usually will incur larger losses due to drift and diffusion effects accumulated over longer time span. So observations will have different weight or impact on the training process, depends on time spans between observations and test data at time t. I understand that the authors want to evaluate prediction impact by starting CSDE from each observation on the test data. However, I am wondering whether it is the best way to define forward loss. Can we just calculate the forward loss between any two consecutive or adjacent observations y? By this way, we can still make full use of all observations to train control agents.

4. Eq. (10), the authors jump to FBSDEs too quickly and it is hard for readers not familiar with this topic to understand. A diagram showing correspondence between forward/backward stochastic processes X/Z and related functions will help readers a lot.

5. Time series data usually show characteristics like level, trend, cycles and holiday/promotion effects. Also in many real applications, time series data come with covariates which may be very predictive for target variables. State-of-the-art algorithms for time series prediction like DeepAR, NBeats, Prophet and Transformer, to name few, take some of the factors into account in the modeling process. It would be interesting to see discussions on how the proposed method handles these factors. Is it possible that different control agents are trained to be experts specialized on different patterns (e.g. trend and cycles) ?

6. In the experiment, an ablation study on the MFcond loss and MBcond loss is encouraged, e.g., use only MFcond loss or MBcond loss to evaluate the algorithm.  It will help the readers to understand how important each loss is.

7. On page 4, in the paragraph 2) Theoretical Optimality, the authors should point out the regularity conditions for the function $V(\cdot)$ while using the Verification Theorem or specify the regularity conditions on the function  $l(\cdot, \cdot)$ and $$\Psi(\cdot), and I think most of the commonly used metric functions in machine learning theory would confirm enough regularity.

8. In Section A.1, I cannot see the necessity to mention the reverse SDE, which has nothing to do with the stochastic control theory the authors referred to in the work. And I think the comparison between the reverse SDE and BSDE is not a related work, I cannot see the motivations or ideas they provide in the work.

Minor issues:

1. Sec. 3.1, define or list reference of "controled F_t – adapted process".

2. Sec. 3.1, define variables d and m.
 Following equation (1), the definitions of the coefficient functions b,\sigma,\alpha^i have typos,
should be written as $b:[0,T]\times \mathbb{R}^d\times\mathbb{A}\to \mathbb{R}^{d}$,  $\sigma:[0,T]\times \mathbb{R}^d\times\mathbb{A}\to \mathbb{R}^{d\times d}$ and $\alpha^i:[0,T]\times \mathbb{R}^d\times\mathbb{R}^m\to \mathbb{R}^n$. And here the admissible control set  $\mathbb{A}$ is not clarified until I read the appendix.

3. Page 3, first line, what is k?

4. Eq. (3), T is replaced by u in the integration. Please explain the replacement. Also I think J(t, X_t^{\alpha}) should be j(T, X_t^{\alpha}) .

5. Eq. (6), there is no y(.) shown in the equation.

6. Eq. (9), what is s in the equation and what is the difference between s and t_s?

7. In the last paragraph on page 2, the 'sub-intervals' of the ordered times $\mathbb{T}$ is confusion. I guess the authors mean: the closed sub-intervals without intersection interiors, or just $[t_k,t_{k+1}], 1\le k\le N$.

8. Another typos in line on page 3: in the expression $w_r(s) = \mathbf{1}_{t\le s\le u}$,  $t, u $ should belong to $\mathbb{T}$, and $1\le k \le N$ should be removed.

9. In Definition 1, the stopping time $\tau_s$ is not well-defined, and I guess the author means : $\tau_s := \inf_{t}\{t>s: l(t,\mathcal{T}_{s,t}^\alpha)>\epsilon\}\bigwedge t$, and the parameter $\epsilon$ should be mentioned in the definition.

10. On page 7, in Section 4, in Table 1, what does $$\Sigma^i mean? Does the Vanilla case indicate that the diffusion term has the same form as the drift term and depends on the controls as well?

**Summary Of The Paper:**

The authors propose a novel method to model stochastic dynamic of complex time series. They build connections between neural SDE and stochastic optimal control theory. By using the proposed MFcond and MBcond losses, they can train control agents to learn dynamics of time series more accurately than existing methods.

**Summary Of The Review:**

The authors propose  a new method for learning dynamics from continuous-time data using neural Markov controlled SDE. It connects stochastic optimal control theory and deep learning which is pretty novel in my opinion. So my recommendation is "accept".

---

> ### Author Response · Authors · 2021-11-13
> **Response to reviewer #WGcS (1/3)**
>
> $\textbf{Q1. }$ Motivation of temporal privacy is not clear. For each agent, it seems that it will only be valid for a specific time interval. The authors may want to explain when a control agent will kick in and when it will be invalid. At the time a control agent kicks in, will it cause a jump of the stochastic process, and is it still a Markov process in this case?
>
> $\textbf{A1. }$
> (1) Motivation of TP: The motivation of temporal privacy is to develop efficient SDE-based models in the context of Markov dynamic programming with the deep-learning framework by decomposing the original problem into several smaller problems.
>
> (2) Potential Risks of Jump: We agree with the reviewer that there exist potential risks of discontinuity at each temporal contact point between control agents if the smoothness/regularity at the finite number of contact points is not guaranteed. However, as our MFcond loss and its corresponding inference procedure take a conditionally averaged process as shown in (21), the total effect of jumps is canceled even if there exists some discontinuity in our forward/backward stochastic dynamics. In practice, to deal with such problems, each control agent shares a small number of temporal states (maximally two points, Appendix  (A.6)) for smooth transitions of stochastic trajectories. While the number of shared temporal states is negligible, temporal privacy is still valid in this case. In all of our experimental results, we have observed no such discontinuity. The toy example in Appendix (A.9) shows some visualization of smooth transitions of the proposed stochastic dynamics.
>
> (3) Markov Property: As the "jump" property is actually one of the main topics deeply dealt with in the analysis of the Markov process, we are happy to discuss the fundamental issues pointed out by the reviewer. We answer the question from the following perspective (Please refer to the additional comments in Appendix (A.6):
>
> The magnitude of jump at the contact point between two control agents is estimated in the context of the Markovian property as shown in the inequality equation (35) in Appendix (A.6), the Markovian property is still preserved, and the magnitude of jumps is controlled by the Lipschitzness of drift/diffusion functions. From a probabilistic point of view, the set of contact points may be regarded as measure-zero, and the probabilistic evaluation is not changed. We believe that, if the stochastic jumps are controlled adequately by extending our CSDE to the general Levy-Ito process (which is also the Markovian) that we have introduced in Appendix (A.10), then the representability of CSDE will be remarkably increased.
>
> $\textbf{Q2. }$ Eq. (4), why the expected terminal loss $J$ given $X_t^{\alpha}$ is optimized using $\alpha^{-r}$? Due to causality, I think it should only depend on partial control agents which are valid after time u.
>
> $\textbf{A2. }$ Due to the property of temporal privacy function, the minimization term with respect to $\alpha^{(-r)}$ in $(B')$ uniformly vanishes if we assume the single temporal privacy function on interval $[t, u]$. In this case, the second term vanishes, and the expectation is only taken in $(B)$. A detailed example is added in Appendix (A.8).

---

> ### Author Response · Authors · 2021-11-13
> **Response to reviewer #WGcS (2/3)**
>
> $\textbf{Q3. }$ Page 4, Definition 1. For each observation y, the dependent forward loss is calculated till time t as shown in the integration. So observations sampled at the time far away from t usually will incur larger losses due to drift and diffusion effects accumulated over longer time span. So observations will have different weight or impact on the training process, depends on time spans between observations and test data at time t. I understand that the authors want to evaluate prediction impact by starting CSDE from each observation on the test data. However, I am wondering whether it is the best way to define forward loss. Can we just calculate the forward loss between any two consecutive or adjacent observations y? By this way, we can still make full use of all observations to train control agents.
>
> $\textbf{A3. }$ As the reviewer suggested, several variants of our method can be also defined for different datasets and applications. For example, we can only take $p$-number of past temporal states to predict future estimation so that the computational burden is somewhat reduced. In this case, the fundamental idea is similar to the $AR(p)$ auto-regressive model. This is a very thoughtful suggestion for our future work, and we believe that our method also benefits from specific applications such as time-series prediction in the online environment and long-term data prediction.
>
> However, we like to point out that there is a clear trade-off between computational complexity and the performance in our method, as the reviewer already recognized. Specifically, as we have investigated in Appendix (A.3), considering multiple conditions (high $p$) leads to the enlarged filtration of the probability space, and the proposed CSDE can possibly induce better performance. Thus, reducing $p$ in our method may lead to the degradation of performance. In experiments, we have observed that the total amount of complexity of our method is comparable to that of state-of-the-art models such as ODE-RNN.
>
> $\textbf{Q4. }$ Eq. (10), the authors jump to FBSDEs too quickly and it is hard for readers not familiar with this topic to understand. A diagram showing correspondence between forward/backward stochastic processes X/Z and related functions will help readers a lot.
>
> $\textbf{A4. }$ We added a clear explanation to introduce the FBSDEs and their main ideas in Section (3.4).
>
> $\textbf{Q6. }$ In the experiment, an ablation study on the MFcond loss and MBcond loss is encouraged, e.g., use only MFcond loss or MBcond loss to evaluate the algorithm. It will help the readers to understand how important each loss is.
>
> $\textbf{A6. }$ As the reviewer suggested, we added the ablation study on the MFcond/MBcond losses in Appendix (A.7). The additional ablation study shows detailed empirical advantages/effects of the proposed losses.
>
> $\textbf{Q7. }$ On page 4, in paragraph 2) Theoretical Optimality, the authors should point out the regularity conditions for the function $V$ while using the Verification Theorem or specify the regularity conditions on the function $l$ and $\Psi(\cdot)$, and I think most of the commonly used metric functions in machine learning theory would confirm enough regularity.
>
> $\textbf{A7. }$ As the regularity condition for the functions is crucial for well-definedness of the Hamiltonian system, we specified the regularity conditions and uniform lipschitzness of functions $J, V, b^i, \partial_x b^i, \partial^2_x b^i, \partial_x \sigma^i, \partial^2_x \sigma^i, \Psi, l$ in Appendix  (A.2) and (A.4).
>
> $\textbf{Q8. }$ In Section A.1, I cannot see the necessity to mention the reverse SDE, which has nothing to do with the stochastic control theory the authors referred to in the work. And I think the comparison between the reverse SDE and BSDE is not a related work, I cannot see the motivations or ideas they provide in the work.
>
> $\textbf{A8. }$ We thought that there is a possible confusion for some readers to understand the context of our method due to the similarity of names (i.e., reverse/backward SDEs). While the reverse SDE is currently drawing the broad attention of AI/ML researchers, we believe that emphasizing the technical/mathematical difference can help identify our method. Additionally, to clarify possible confusions, we will also comment/emphasize the difference between our proposed method and neural controlled differential equation (Kidger et al. 2020), which also shares a similar name to ours.
>
> $\textbf{Q9. }$ Eq. (9), what is $s$ in the equation and what is the difference between $s$ and $t_s$?
>
> $\textbf{A9. }$ Let us remind that the set $I(t_m, t_k)$ indicates the set of time indices from $t_m$ to $t_k$. In this case, the value $s$ denotes the number of index in the set $I$. Using this, $t_s$ means that $s$-th time stamp, and $\hat{y}_{t_s}$ in (9) indicates the observed data at time $t_s$.

---

> ### Author Response · Authors · 2021-11-13
> **Response to reviewer #WGcS (3/3)**
>
> $\textbf{Q10. }$ On page 7, in Section 4, in Table 1, what does $\Sigma^i$ mean? Does the Vanilla case indicate that the diffusion term has the same form as the drift term and depends on the controls as well?
>
> $\textbf{A10. }$ It is a typo. The notation $\Sigma^i$ is now corrected to $\sigma^i$ in the table, and it is moved to Table 4 in Appendix (A.6).
>
> $\textbf{Q11.}$ Eq. (3), T is replaced by u in the integration. Please explain the replacement. Also I think J(t, X_t^{\alpha}) should be j(T, X_t^{\alpha}).
>
> $\textbf{A11.}$ We followed the identical notation/rule in Theorem 2.1 (Yong & Zhou, 1999). (Bellman's optimal principle with Markov DP)
>
> Other minor issues are fully considered in the revised version of draft. We are infinitely grateful for the reviewer's awareness of theoretical contributions, and careful suggestions.

---

### Official Review · Reviewer_hDHJ · 2021-11-01

**Correctness:** 3
**Technical Novelty And Significance:** 3
**Empirical Novelty And Significance:** Not applicable
**Recommendation:** 6
**Confidence:** 2

**Main Review:**

The paper seems to be well-structured and the math seems solid. The proposed loss function MFcond is a principled way to incorporate the intermediate observations into the dynamics. The experiments shows significant gains over several state-of-the-art baselines. I have a few comments and concerns:
 - The motivation behind modelling a time series such as stock price as controlled by multiple agent is not clear.
 - It is not clear to me whether multiple agents can be active at the same time. From the definition and the separation of B and B' in equation 4, it seems that the attention interval of two different agents are exclusive, but the example in figure 4 seems to show that agents control overlapped intervals.
 - This paper does not clearly define how the attention of each agent is set for empirical datasets. From the context it seems that each agent controls one interval between two empirical observations, but this need to be explicitly defined.
 - The figure 1-(b) shows a failure case where all simulated trajectories are far from the ground truth, yet the "averaged" prediction operator $\mathcal{T}_{s,t}^{\alpha}$ is close to the ground truth. Doesn't this mean that l2 distance between the ground truth and the mean predictor is a poor loss function?
 - In figure 1 the observation is plotted in a solid black line. I think scatter plot is more appropriate since the observations are at discrete time.

**Summary Of The Paper:**

This paper proposes to model a time series directly with a neural controlled stochastic differential equation with multiple control agents. It introduces a concept of temporal privacy, which defines the attention of each control agent to be a certain time interval. Then the authors introduce Markov dynamic programming to efficiently minimize a loss function defined in terms of trajectory of the stochastic differential equations. In order to make the proposed temporal privacy work, the authors proposed two loss function: 1) MFcond minimizes the incoherence between the future estimates starting at any intermediate time stamp, 2) MBcond is to ensure the theoretical optimality of the control agents.

**Summary Of The Review:**

This paper proposes some novel ideas that works well in practice. I don't fully understand this paper, but I feel it deserve a broader audience.

---

> ### Author Response · Authors · 2021-11-14
> **Response to reviewer #hDHJ**
>
> $\textbf{Q1. }$ The motivation behind modelling a time series such as stock price as controlled by multiple agent is not clear.
>
> $\textbf{A1. }$ (1) Modelling stock market data as the CSDEs: In financial engineering, the stochastic differential equation is the fundamental mathematical object deeply investigated/applied in a real-world situation to deal with stock market data. Our method follows their philosophy to model the time-series data that may contain the volatility or unknown stochastic perturbations that can not be easily captured by the deterministic dynamical models such as ODEs. Our primary objective in this paper is to \textbf{control the SDE models with neural network} (a) to delicately mimic time-series data (historical stock prices), (b) to predict future value (future stock price).
>
> (2) The motivation of multi-agent CSDE: While the prior works focus on the combination of ODE/SDE with the additional RNN/LSTMs, the total amount of complexity is much higher than that of ours, and the power of the SDE is suppressed in their methods. In this paper, we have built an efficient model called CSDE-TP (with multi-agents), which maximizes the learnability/power of SDE model to generate/predict time-series dataset. Unfortunately, training naive CSDE (with single-agent) is not the answer to efficiently solve the optimization problem in equation (4) as standard CSDE is inefficient and produces poor test results, as shown in black dotted line Figure 2-(a). Thus, we tackle solving this problem by combining two ideas. First, as shown in our CSDE-TP in equations (1) and (2), the stochastic trajectories $X_t$ are propagated by summing each agent's attention (temporal privacy) on their specialized sub-intervals. Second, We apply the Markov dynamic programming principles to our CSDE-TP. As this harmonizes with the first idea, the original stochastic optimal control problem (A) in equation (4) is naturally decomposed into two sub-problems (B) and (B'), and each control agent can efficiently compute the gradient descent in their specialized intervals. In Fig 2-(a), the blue, black lines show the efficiency of the CSDE-TP (multi-agents) compared to naive CSDE (single agent).
>
> $\textbf{Q2. }$ It is not clear to me whether multiple agents can be active at the same time. From the definition and the separation of B and B' in equation 4, it seems that the attention interval of two different agents are exclusive, but the example in figure 4 seems to show that agents control overlapped intervals.
>
> $\textbf{A2. }$  In principle, each control agent does not share any temporal states in our method. Unfortunately, there may be a minor theoretical issue when we completely isolate temporal states between control agents. Specifically, as pointed out by other reviewer #WGcS, TP induces some potential risk that causes jumps between contact points of control agents if the dataset contains large discontinuities. In practice, setting a few sharing points prevents such jumps and produces smooth transitions. As the sharing points are negligible in our experiments, we believe that the role of temporal privacy is still valid in our method.
>
> $\textbf{Q3. }$ This paper does not clearly define how the attention of each agent is set for empirical datasets. From the context, it seems that each agent controls one interval between two empirical observations, but this needs to be explicitly defined.
>
> $\textbf{A3. }$ We added additional comments on this point in our Appendix (A.6).
>
> $\textbf{Q4. }$ Figure 1-(b) shows a failure case where all simulated trajectories are far from the ground truth, yet the "averaged" prediction operator $\mathcal{T}_{s,t}^{\alpha}$ is close to the ground truth. Doesn't this mean that l2 distance between the ground truth and the mean predictor is a poor loss function?
>
> $\textbf{A4. }$ Figure 1-(b) shows the result of the failure case, we intentionally demonstrated the failure case to visually show the fundamental idea of the proposed inference procedure. For example, even if every single prediction starting at different initial states $t_{m_1}, \cdots, t_{m_3}$, the "averaged" predictor $\mathcal{T}$ (We call it $n$-parameter martingale in Appendix (A.3)) shows the relatively accurate results as it utilizes multiple information to predict future evaluations. In our real experiments, the averaged predictor produces accurate results, as shown in the toy example of Appendix (A.9).

---

### Official Review · Reviewer_poqQ · 2021-11-01

**Correctness:** 4
**Technical Novelty And Significance:** 3
**Empirical Novelty And Significance:** 3
**Recommendation:** 3
**Confidence:** 4

**Main Review:**

Strengths:
+ The paper concerns a timely topic. The stochastic view of the continuous-time systems makes it even more interesting as ordinary differential equations based approaches dominate the recent literature for the time being.
+ The empirical findings are strong. The model seems to excel among state-of-the-art continuous-time methods in standard benchmarks.
+ We also empirically observe that the proposed MFcond significantly boosts the accuracy. Note that this comparison is made against the vanilla optimization objective (6). Since it is known that training continuous-time systems with long sequences suffer from optimization issues; thus, it would be more appropriate to consider an additional baseline in which the training is performed using subsequences (i.e., minibatches along the time dimension).

Weaknesses:
- Since the model addresses difficult sub-problems along the way, these sub-problems and appearing hyperparameters require a more in-depth investigation/explanation. In turn, this would allow us to appreciate the methodology and different choices the authors have made. This can be done, for instance, by simple empirical evaluations (as in Fig 4.4a) or by contrasting the proposed strategies against any simpler, alternative techniques. Unfortunately, as such, I'm not able to judge how much is gained by the proposed complicated and somewhat demanding procedures. Below are a few simple example questions:
    - What happens when we discard the MFcond? More generally, what is the role of $\gamma$ and how did you set it $\gamma=0.95$?
    - How do you pick $u$ in (3)? Does the optimization become more difficult/easy with $u$ approaching $t$?
    - How does the value for $r$ affect the entire routine defined in (4)? Did you randomly pick $r$
    - How does $\epsilon$ affect the algorithm? Why is it set as in Alg.1?
    - What if we simply optimize for shorter sequences instead of (6)?
    - Setting the initial value to an observation (as in (6),(7)) may deteriorate the performance if the data is noisy. Is not this an issue after all?
    - What is the justification for replacing $Z_t$ with $Z_t^\alpha$? Since this directly violates the "optimal agent" requirement of (11), this particular choice needs to be carefully explained, possibly paired with a simple numerical illustration showing how it affects (10).
- Writing and notation should be significantly improved. Below are concrete suggestions:
    - Since (1) involves several terms, more verbal explanation would help the reader very much. Also, short explanations on "Markov closed-loop feedback control" and "$\mathcal{F}_t$-adapted process" are needed.
    - A cartoon drawing (or anything similar to Figure 1) of Sec 3.1 would be nice.
    - What are the expectations with respect to in (3)?
    - A verbal explanation of (7) would be very useful.
    - The caption of Fig 1 should be improved. It is not clear what "other trajectories", "it", and "empty parts" refer to.
    - Which distribution is the expectation in (8) with respect to? Does not $\mathbf{y}(\cdot)$ refer to data trajectories?
    - It would be much easier to follow if the paragraph above "Network inference" is given much earlier than equations.
    - The first paragraph in page 6 is very long and difficult to follow.
    - Sec3.4 could start with a reminder of MBcond as it is not related to the methodology described in Sec 3.3.
    - What exactly does "cancels the effect of martingales in the diffusion term" mean?
    - Writing (12) in terms of differentials (as in (10)) instead of an integral would help the reader to contrast (10) and (12).
    - What are the inputs to $\mathcal{L}_f$? Just $\alpha$'s as in (14) or $\mathbf{y}(\cdot)$ as well as in (8)? Also, why is the $\alpha^i$ separated from the other $\alpha$'s in (14)?
    - Better titles appearing in Fig 4a legend (maybe reflecting the number of agents) would be nice.

More detailed comments and questions:
- Why are the Q1 and Q2 important? Q1 needs further explanation and Q2 requires an explanation on the deficiencies of neural controlled differential equations (NCDE).
- NCDE should be mentioned in A1 and the proposed method should be motivated in comparison with NCDE.
- What prevents the agents from collapsing to a single mode?
- Why is only the train MSE plotted in Fig 4a? Also, it would be nice to see the model trained until convergence.

**Summary Of The Paper:**

This paper presents a new approach to train stochastic dynamical systems using a set of tools from stochastic optimal control theory. The methodology is built upon _controlled stochastic differential equations_ with multiple control agents that modulate the dynamical evolution. Each agent is deliberately chosen to be "active" in a certain time interval, which leads to so-called "temporally private" dynamics that allow for dynamic programming based optimization. Since the temporal segmentation of optimization objective requires arbitrary intermediate state $X_s$ with $0<s<T$ as initial values, the Markov forward condition (MFcond), which computes some sort of a mean value for future states, is proposed. To further stabilize the training and compensate for the methodology's lacking in theoretical optimality guarantees, the authors augment the original loss with the Markov backward conditional (MBcond) loss. The method is shown to outperform competing continuous-time methods on standard benchmarks.

**Summary Of The Review:**

The paper brings several interesting approaches to address the sub-problems that appear due to the construction of the SDE controlled by multiple agents. As much as I can follow, the equations are accurate and the methodology does not suffer from any theoretical deficiency. On the other hand, this makes it very difficult to evaluate how important different assumptions and building blocks are. As such, the model seems like a magical combination of a number of blocks but it would be much better to isolate each block individually and show how it contributes to the overall performance. Also, I believe the writing should be significantly improved (see my detailed notes in the "main review"). Although the method is interesting and timely, I recommend a reject and re-submission after the above-mentioned points are addressed.

---

> ### Author Response · Authors · 2021-11-12
> **Response to reviewer #poqQ (1/4)**
>
> We gratefully thank the reviewers for thorough feedback, valuable suggestions, and insights. Certain parts of the draft suggested by reviewers are carefully updated.
>
> $\textbf{Q1. }$ What happens when we discard the MFcond? More generally, what is the role ofγand how did you set it γ=.95?
>
> $\textbf{A1. }$ The MFcond loss is the central training objective in our method because it collaborates with the proposed novel inference procedure, and fully exploits the information of complex time-series data. Thus, without the MFcond loss, the proposed CSDE-TP produces poor performances. While it is customary to balance the multiple objective functions with hyper-parameters, the role of $\gamma$ can be understood as a coefficient to balance between two objective functions, (i.e.,  MFcond and MBcond).
>
> If $\gamma = 0$, then the MFcond loss is discarded as the reviewer requested, and the method is trained only with the MBcond loss in this case. In Figure 5-(a) of the revised paper (Appendix (A.7)), the blue line shows the landscape of test MSE for this case. As we aforementioned, a significant performance drop is observed, and the proposed CSDE-TP hardly predicts the time-series data.
>
> If $\gamma = 1$, then the MBcond loss is discarded, and the proposed method is trained only with the MFcond loss. In Figure 5-(a) of the revised paper (Appendix (A.7)), the red line shows the landscape of test MSE for this case. As the MBcond loss plays an auxiliary role in our method (e.g.,  regularization effect), it is clear that MBcond loss yields the performance gain, as shown in Figure 5-(a).
>
> We empirically tuned hyper-parameter $\gamma$ between $0 \leq \gamma \leq 1$ to produce the best performance. In all experiments, this value is fixed to $0.95$.
>
> $\textbf{Q2. }$ How do you pick $u$ in (3)? Does the optimization become more difficult/easy with $u$ approaching $t$? How does the value for $r$ affect the entire routine defined in (4)? Did you randomly pick $r$?
>
> $\textbf{A2. }$ The values $u$ and $r$ are not determined values. Rather, $u$ and $r$ in equations (3) and (4) represent the auxiliary indices for time $(0 \leq u \leq T)$ and control agents $(1 \leq r \leq M)$, respectively.
>
> The role of $u$ in equations (3) and (4) is replaced to $u$ and $u'$ in $\mathbf{(P3)}$ in Appendix (A.8), and replaced to $(r * \frac{T - t}{M})$ in $\mathbf{(P4)}$ in Appendix (A.8) if the time interval is assumed to be regularly sampled. Similarly, the role of $r$ in the equations (3) and (4) is replaced to $r$ and $r'$ in $\mathbf{(P3)}$. In all cases, we believe that the role of $u$ and $r$ is clear in the context.
>
> In our experiments, the number of control agents $M$ is set to $|\mathbb{T}|/2$, where $|\mathbb{T}|$ is the number of temporal states of time-series data. We provided further detailed information on design factors in Appendix (A.6).
>
> $\textbf{Q3. }$ How does $\epsilon$ affect the algorithm? Why is it set as in (Alg.1)?
>
> $\textbf{A3. }$ In Appendix  (A.7), Figure 5-(b) shows the effect of two different strategies for selecting stopping time $\epsilon$. If we select stopping $\epsilon$ randomly as shown in the red line of Figure 5-(b), the network quickly falls into instability, and the proposed model produces poor results. Contrary to the random strategy, the proposed one in (Alg.1) stabilizes the landscape of test MSE, as shown in the black line of Figure 5-(b). We added some comments on the stability of the proposed strategy in equation (35).
>
> $\textbf{Q4. }$. What if we simply optimize for shorter sequences instead of (6)?
>
> $\textbf{A4. }$ Suppose that we slice the sequence into two shorter sub-sequences and optimize the objective function. To generate accurate time-series data, smooth transitions between temporal states are essential for stochastic dynamical models.
> Unfortunately, if we train a pair of sub-sequences in parallel with a batch-wise manner, the endpoint of the first sub-sequence will not be matched to the first point of the second sub-sequence as our model is defined as the solution of SDE. In this case, large jumps appear at every contact point between sub-sequences, and the generated results will not be satisfactory. To prevent such problems, it is beneficial to deal with the entire sequence as a single time-series object.

---

> > ### Author Response · Authors · 2021-11-12
> > **Response to reviewer #poqQ (2/4)**
> >
> > $\textbf{Q5. }$ Setting the initial value to an observation (as in (6),(7)) may deteriorate the performance of the data is noisy. Is not this an issue after all?
> >
> > $\textbf{A5. }$ The stability with noisy initial states is an important/fundamental issue in a complex dynamical system, which has to be deeply dealt with. Unfortunately, to the best of our knowledge, there exists no prior work (ODE/SDE-based models) exploring stability given noisy initial states. However, we have found some connection between the prior work in [A] and the stability of the proposed method. In [A],  the authors explored the Lyapunov exponent of controlled SDE (similar to ours), and their results show that the Lipschitzness of drift and diffusion functions is crucial to stability. Thankfully, in our network structure, as shown in Appendix (A.6), we utilize LipS with functions, which makes our network satisfy the Lipschitzness. Thus, our network produces robust results even if the small noises are injected in the initial states.
> >
> > [A] Lyapunov exponents of controlled SDE’s and stabilizability property: Some examples, Fabien Campillo, Abdoulaye Traore, INRIA, technical report
> >
> > $\textbf{Q6. }$  What is the justification for replacing  $Z_t$  with $Z_t^{\alpha}$? Since this directly violates the "optimal agent" requirement of (11), this particular choice needs to be carefully explained, possibly paired with a simple numerical illustration showing how it affects (10). Writing (12) in terms of differentials (as in (10)) instead of an integral would help the reader to contrast (10) and (12).
> >
> > $\textbf{A6. }$ The optimal FBSDE states the variable $Z_t$ can be written in following form:
> > \begin{equation}
> >     dZ_t = - l(s, X_t^{\alpha_{\star}})dt + \sum_i^M \nabla V(t, X_t^{\alpha_{\star}}) \sigma^i dW_t
> > \end{equation}
> >
> > The equation (12) in the paper can be rewritten as the Ito's differential form as follows:
> > \begin{equation}
> >     dZ^{\alpha}_t = - l(s, X_t^{\alpha})dt + \sum_i^M \nabla J(t, X_t^{\alpha}) \sigma^i dW_t
> > \end{equation}
> >
> > In both equations, there is a major difference between them, i.e.,  $\nabla V$ and $\nabla J$. If we solve the original problem (A) in equation (4), then $\alpha = \alpha_{\star}$, and we can conclude that $V = J$ and $\nabla V = \nabla J$. Thus, approximating $Z_t^{\alpha}$ to $Z_t$  is a $\textbf{mathematically identical reformulation of $J$ in (4)}$ in the main paper. Thus, no conflict occurs in this case. In Appendix (A.4), the detailed procedure to induce $Z$ in (10) from $J$ (4) is presented.
> >
> > The main point in Appendix (A.4) is that we can provide the additional information (conditional expectations) to backward dynamics $Z_t^{\alpha}$ by training the MBcond loss in (13). We believe that the unclarity may come from the integral form of backward dynamics $Z_t^{\alpha}$ in (12). Thus, we will modify the equation (12) to the differential form as suggested by the authors to remove any possible uncleanness.
> >
> > The additional experiment results in the appendix show the specific effect of backward dynamics $Z_t$ with the MBcond loss. We believe that this result will help readers to understand the concept of MBcond loss.
> >
> > $\textbf{Q7. }$ Since (1) involves several terms, more verbal explanation would help the reader very much. Also, short explanations on "Markov closed-loop feedback control" and "$\mathcal{F}_t$-adapted process" are needed.
> >
> > $\textbf{A7. }$ Thanks for your careful suggestion. The short explanations will be added in the revised version as follows:
> >
> > "$\mathcal{F}_t$-adapted process means that the proposed stochastic process $X_t$ is continuously measurable according to the increasing time $0 \leq t \leq T$.
> >
> > "Markov closed-loop feedback control" indicates that the control agents in our proposed controlled SDE only take the current information (Markov property) to propagate the stochastic trajectories.
> >
> > We believe that these definitions enhance the mathematical rigor of the theoretical parts of our paper.

---

> ### Author Response · Authors · 2021-11-12
> **Response to reviewer #poqQ (3/4)**
>
> $\textbf{Q8. }$ Why are the Q1 and Q2 important? Q1 needs further explanation.
>
> $\textbf{A8. }$ While the significant number of prior works focused on the combination of SDE and recurrent models, as we state in the introduction section, even if the SDE model itself is already a powerful probabilistic model, SDE is dealt with simply as a probabilistic part of these combinations. In prior works, the naive objective function is utilized, and the learnability of the SDE-based model is suppressed. Thus, naive approaches lead to the inefficient use of the SDEs.
>
> To overcome these issues, we tackle solving the two major issues as follows:
>
> $\bullet$  Importance of Q1  (Network Efficiency). We alternatively define the stochastic optimal control-theoretic framework and suggest several techniques to improve efficiency. For example, the combination of temporal privacy and Markov dynamic programming framework substantially increases efficiency, as shown in Section 4.4 and Figure 2-(a). This approach makes our model solely propagate complex time-series data without additional networks such as RNNs, and the total computational complexity is considerably reduced.
>
> $\bullet$ Importance of Q2 (Network Learnability). Beyond the mathematical meaning, the objective function in (6) shows the fundamental methodology to train ODE/SDE in existing methods (NODE). For example, the core idea is "For the given single initial state, the ODE/SDE model is trained to generate multiple future states."
>
> This idea causes a problem because the dynamical model only accesses the partial information of the entire time-series data. In this case, there is no guarantee that the ODE/SDE model can generate multiple different time-series data which share identical initial states. In our method, the core idea is "For the given multiple past states, our SDE model is trained to generate multiple future states." This idea is a core part of the MFcond loss in equation (7). As expected, multiple conditions boost performance, as shown in Figure 2-(b).
>
> $\textbf{Q9. }$  Q2 requires an explanation of the deficiencies of neural controlled differential equations (NCDE). NCDE should be mentioned in (A1) and the proposed method should be motivated in comparison with NCDE.
>
> $\textbf{A9. }$ The neural controlled differential equations (NCDE, Kidger, et al. 2020), which share a similar name with ours, suggest a different mathematical theory. Specifically, their method focuses on applying $\textbf{rough-path analysis theory}$ (they cited a series of works done by T. J. Lyons). In contrast, the neural controlled stochastic differential equation (CSDE) suggested in our paper focuses on applying $\textbf{stochastic optimal control theory}$, which lies in a different mathematical context. The authors of NCDE also pointed out the potential confusion of two other theories that share similar names. With all due respect, we hardly think that the CSDE should be motivated by NCDE. As future readers may confuse this point, we will add comments in Section 2 for clarity.

---

> ### Author Response · Authors · 2021-11-12
> **Response to reviewer #poqQ (4/4)**
>
> $\textbf{Q10. }$ What are the expectations with respect to in (3)?
>
> $\textbf{A10. }$ The conditional expectation is taken to the probability measure $\mathbb{P}$ that we have defined in Appendix (A.2). As the stochastic process $X_t$ is the Markov process, the second equality in (3) holds by Definition 3.
>
> $\textbf{Q11. }$ Which distribution is the expectation in (8) with respect to? Does not $y_{(\cdot)}$ refer to data trajectories?
>
> $\textbf{A11. }$ As equation (7) is presented as conditional expectations given each temporal state of data trajectories, the expectation in (8) represents the expectation to time-series data.
>
> $\textbf{Q12. }$ Other questions/suggestions
>
> $\textbf{A12. }$ The other questions/suggestions (e.g., notations and writing) by the reviewer are fully considered in the revised version of the paper.

---

### Official Review · Reviewer_NnV6 · 2021-11-02

**Correctness:** 4
**Technical Novelty And Significance:** 4
**Empirical Novelty And Significance:** 4
**Recommendation:** 8
**Confidence:** 4

**Main Review:**

This paper formulates a stochastic differential equation that is controlled by $M$ external agents over compact, non-overlapping time intervals. Due to the lack of overlap between the agents, there is only one agent controlling the dynamics at each sub-interval, a notion that the authors call "temporal privacy". The goal, as in works using the neural SDE framework, is to learn stochastic time-series data using this model. Using a stochastic optimal control formulation, the authors use this notion of temporal privacy to decompose the optimization into sub-problems which allows for separate optimization of each agent.

However, carrying out the training of this model is nontrivial. To train, the authors first use a loss function that employs random stopping times to efficiently use the data $\boldsymbol{y}$. The inference then proceeds similarly, propagating the sampled points according to the learned dynamics and subsequently averaging the results. A second loss that describes the evolution of the loss backward in time is used to augment this first loss. I think the connection of the second loss (MBcond) to the nonlinear Feynman-Kac theorem could be better explained in the main text.

The authors carry out several numerical experiments and achieve impressive results. In all cases the CSDE-TP significantly outperforms alternative strategies.

Minor comments:

Figure 1 caption: "It computes the ..." what is "it"?

Typo: "is shoen in"

"main idea is ... to minimize incoherence" what does this mean? can this notion of incoherence be made more precise?

The phrase "rich information" comes up a few times. Perhaps it is better to simply say "information"





**Summary Of The Paper:**

This paper presents a framework for learning a stochastic dynamics from observed trajectories. As opposed to conventional strategies based on recurrent neural networks, the procedure learns a model of the stochastic dynamics in real space. Secondly, in contrast to recent approaches based on neural ODEs / SDEs, the training is temporally localized through windowing functions that the authors refer to as temporal privacy. The main technical innovations involve the development of new loss functions that make learning with temporal privacy tractable.

**Summary Of The Review:**

The author introduce an original and highly effective optimization scheme for controlled SDEs for modeling stochastic dynamics. While there is not a thorough analysis of the algorithm (aside from a discussion of the theoretical optimality in the appendix) the experiments amply demonstrate that the approach is productive.

---

> ### Author Response · Authors · 2021-11-13
> **Response to reviewer #NnV6**
>
> We are infinitely grateful for the reviewer's awareness of theoretical contributions. The draft is revised by considering the reviewer's valuable suggestions.
>
> $\textbf{Q1. }$ I think the connection of the second loss (MBcond) to the nonlinear Feynman-Kac theorem could be better explained in the main text.
>
> $\textbf{A1. }$ We added additional comments on bridging the concept of the proposed MBcond loss with the optimality of CSDE in Section 3.4. We emphasized the optimality related to FBSDE is obtained from the connecting non-linear Feynmann-Kac formula to the HJB equation.
>
> $\textbf{Q2. }$ On the minor comments.
>
> $\textbf{A2. }$ Regarding the minor issues pointed out by the reviewer, we modified our draft as follows:
>
> (1) The caption of Figure 1: "It computes the ..." $\rightarrow$ "The MFcond loss estimates the .."
>
> (2) Typo: "is shoen in" $\rightarrow$ "is shown in"
>
> (3) The sentence is modified to clarify its meaning: "main idea is ... to minimize incoherence" $\rightarrow$ "main idea is ... to minimize the differences"
>
> (4) "rich information" $\rightarrow$ "information"

---

### Decision · Program_Chairs · 2022-01-20

**Decision:**

Accept (Poster)

**Comment:**

The authors propose to combine ideas from SDEs and time series modeling with stochastic optimal control to present a framework for modeling continuous-time stochastic dynamics. The reviewers are in agreement that there are several good ideas presented here and that the interface of the perspectives the authors combine toward their proposed framework is an interesting one to explore. One referee mentions valid concerns in confusing points of the details in the presentation, and the positive reviewers echoed these concerns. In particular, more details and clearer exposition are needed for the decomposition into the subproblems and the problem of many hyper parameters. Nonetheless, my overall impression after a careful read of the paper and discussion is that these concerns are addressable and are to a degree ameliorated by the author response, and that they may be viewed as limitations outweighed by the merits of the novel ideas presented here. I emphasize that all reviewers were surprisingly consistent in their assessment of the shortcomings, and I encourage the authors to take these constructive criticisms seriously in preparing a final version of this paper.